# Interpretable spatially aware dimension reduction of spatial transcriptomics with STAMP

Chengwei Zhong[1,2], Kok Siong Ang[1,2] & Jinmiao Chen ●[1,2,3,4] ✉

Spatial transcriptomics produces high-dimensional gene expression measurements with spatial context. Obtaining a biologically meaningful low-dimensional representation of such data is crucial for effective interpretation and downstream analysis. Here, we present Spatial Transcriptomics Analysis with topic Modeling to uncover spatial Patterns (STAMP), an interpretable spatially aware dimension reduction method built on a deep generative model that returns biologically relevant, low-dimensional spatial topics and associated gene modules. STAMP can analyze data ranging from a single section to multiple sections and from different technologies to time-series data, returning topics matching known biological domains and associated gene modules containing established markers highly ranked within. In a lung cancer sample, STAMP delineated cell states with supporting markers at a higher resolution than the original annotation and uncovered cancer-associated fibroblasts concentrated on the tumor edge's exterior. In time-series data of mouse embryonic development, STAMP disentangled the erythro-myeloid hematopoiesis and hepatocytes developmental trajectories within the liver. STAMP is highly scalable and can handle more than 500,000 cells.

Spatial transcriptomics is an essential experimental technique for exploring tissue architecture as it captures gene expression profiles while retaining their spatial context[1]. Consequently, spatially aware analyses are required to incorporate gene expression and spatial information to fully exploit such data. In commonly employed workflows, dimension reduction forms the initial processing step, which is particularly important as its accuracy in capturing the relevant data variability impacts data visualization and downstream analyses such as clustering to identify biologically relevant spatial domains or cell types. Thereafter, differential expression analysis typically follows to enable annotation. Alternatively, interpretable dimension reduction is an attractive option that exploits the computed contribution of each gene

to the low-dimensional embeddings. By visualizing such embeddings on the spatial coordinates and examining the contributing input genes, one can decipher anatomical regions or cell types without clustering and differential expression analysis. This also gives greater confidence in the biological relevance of the reduced dimensions compared to black-box modeling approaches.

Classical dimensional reduction methods such as principal-component analysis (PCA), non-negative matrix factorization (NMF)[2] and latent Dirichlet allocation (LDA)[3] are often used in single-cell analysis[4–6]. They return interpretable embeddings but may lack expressivity and do not incorporate spatial information. Alternatively, autoencoder (AE) approaches such as linearly decoded variational

[1]Bioinformatics Institute (BII), Agency for Science, Technology and Research (A*STAR), Singapore, Singapore. [2]Singapore Immunology Network (SIgN), Agency for Science, Technology and Research (A*STAR), Singapore, Singapore. [3]Centre for Computational Biology and Program in Cancer and Stem Cell Biology, Duke-NUS Medical School, Singapore, Singapore. [4]Immunology Translational Research Program, Department of Microbiology and Immunology, Yong Loo Lin School of Medicine, National University of Singapore (NUS), Singapore, Singapore. ✉e-mail: micchenj@nus.edu.sg

autoencoder (LDVAE)[7] combine nonlinear encoders with linear decoders to balance expressive power and interpretability, but also do not incorporate spatial information. Spatial transcriptomics specific methods like SpaGCN[8], BASS[9], BayesSpace[10] and GraphST[11] conversely employ sophisticated models such as graph neural networks or the Potts model to incorporate spatial information. These methods offer flexibility and can jointly analyze omics data with spatial information; however, the embeddings returned by these methods are usually interpreted post hoc, with clustering and differential expression analysis. Current interpretable methods that incorporate spatial information into dimension reduction employ Gaussian processes or hidden Markov random fields, as exemplified by methods like MEFISTO[12], non-negative spatial factorization hybrid (NSFH)[13] and spatial identification of cells using matrix factorization (SpiceMix)[14]. They provide spatially aware interpretable dimensional reduction but are computationally costly.

Here, we present STAMP for interpretable spatially aware dimension reduction. STAMP builds upon the deep generative model prodLDA[15], which ensures scalability through auto-encoding and black-box variational inference[16,17]. STAMP uses a simplified graph convolution network[18,19] as an inference network, allowing spatial information to be incorporated at a small increase in computational cost. STAMP outputs a latent representation consisting of spatially organized topics with associated gene modules containing genes ranked by their contribution to the topic. This explicitly links the importance of each gene to a topic, contributing to interpretability. If desired, further analysis such as gene set enrichment or pathway analysis can assign further biological meanings to each topic. STAMP computes a topic proportion score for each cell and each topic, with the proportions summing to 1 within each cell. For cells with a dominant topic, the interpretation associated with that dominant topic can also be assigned. We also introduce structured sparsity-inducing priors[20,21] on the gene modules learned such that the modules are both sparse across topics and within topics. We tested STAMP on several datasets and showed that the learned topics correspond to known anatomical structures in tissues. This obviates the need for an additional clustering step. Furthermore, the relevant marker genes were highly ranked in the accompanying gene modules, highlighting STAMP's interpretability and offering an alternative to differential expression analysis.

We benchmarked STAMP against other dimension reduction methods that output non-negative latent embeddings and their corresponding gene modules, and showed that STAMP performed favorably in identifying biologically relevant domains and their gene modules. In addition, STAMP also outperformed other methods in terms of scalability. Finally, STAMP is the first topic modeling method that can capture shared topics across time-series spatial transcriptomic data. With a mouse embryonic development dataset, STAMP revealed spatiotemporally linked topics and their associated gene modules. These topics matched known biological structures and the associated gene modules tracked changes in the contributing genes across time.

## Results

### Workflow of STAMP

STAMP combines topic modeling with deep generative models, thus inheriting the benefits of interpretability and scalability. It takes in gene expression values with their spatial context and outputs explainable latent topics with associated gene modules (Fig. 1a,b). Each associated gene has a gene module score that denotes its contribution to the topic. In its most basic form that handles a single sample, STAMP models the observed expression $x_{ng}$ of gene $g$ in cell $n$ as a sample drawn from a Gamma Poisson distribution determined by a combination of the latent topic $z_{nk}$ (the proportion of topic $k$ in cell $n$), gene module $w_{kg}$ (the contribution of gene $g$ to topic $k$), background residual $r_g$ and dispersion $\alpha_g$. To promote structured sparsity in the gene modules $w_{kg}$, we utilize a structured regularized horseshoe prior[22,23]. This prior ensures that each gene is involved in only a subset of topics and each topic involves

only a limited number of genes in the associated gene module, which facilitates biological interpretation of the modules and provides robustness to these modules (a detailed description of the prior is given in the Methods section). For handling multiple samples, we add a batch correction term, $w^{batch}$, to capture batch related variations present in the data. Lastly, we expand STAMP to handle time-series data, allowing the gene modules to vary across different time points by imposing a Gaussian process prior with a Matern kernel. To incorporate spatial information into the model, we use a simplified graph convolutional network (SGCN) as our inference network that takes in gene expression and an adjacency matrix built from the spatial locations (Fig. 1a). The whole model is trained end to end with black-box variational inference by maximizing the evidence lower bound (ELBO). A detailed description of the model can be found in the Methods and Supplementary Note 1.

We first validated STAMP's ability to recover different layers of patterns using a simulated dataset. Following the approach of Townes et al.[13], we generated data with five overlapping patterns (Extended Data Fig. 1a), where each spot could be associated with multiple patterns. STAMP was able to cleanly recover the five patterns (Extended Data Fig. 1a,b). We also evaluated other methods, including Leiden[24] clustering, spatially aware dimension reduction with STAGATE[25], GraphST[11] and SpatialPCA[26], followed by $k$-means clustering. None of these clustering-based analyses was able to correctly recover all the patterns (Extended Data Fig. 1c). This demonstrates the advantage of STAMP's topic-modeling approach over clustering algorithms in deconvoluting spatially overlapping patterns. Unlike traditional clustering, which assigns each spot to a single cluster, topic modeling provides a probabilistic topic composition for each spot, allowing for multiple identities (topics) to be represented. Additionally, we tested cluster-free methods, namely LDA, LDVAE, NMF, NSFH and SpiceMix. While these methods outperformed clustering approaches, they were less effective than STAMP (Supplementary Note 2).

### STAMP reveals spatial domains of mouse hippocampus

Next, we demonstrated STAMP's ability to identify biologically relevant topics and gene modules of known tissue structures. We applied STAMP to a mouse hippocampus dataset of 39,220 spots generated using Slide-seq V2 (ref. 27). Using the Allen Brain Atlas as the ground truth for anatomical regions[11,28] (Fig. 2a), we compared the performance of STAMP and five other methods, NMF[2], LDA[3], LDVAE[7] and two recent methods developed for spatial transcriptomics, NSFH[13] and SpiceMix[14]. All these methods output non-negative latent embeddings and their corresponding gene modules.

We first quantitated the performance of STAMP and competing methods in terms of the topic associated gene modules (Fig. 2b). We employed two commonly used metrics in topic modeling literature, module coherence[29] and module diversity[30]. Module coherence measures the degree of coexpression among the top-ranking genes within a module, whereas module diversity measures the uniqueness of a gene module. STAMP scored the highest in both metrics with a median module coherence of 0.162 and module diversity of 0.9. All other methods scored much poorer in the coherence metric with NMF being second (0.127). For diversity, LDVAE scored second (0.87) while being much poorer at module coherence (0.11). LDA was the weakest performer, scoring last in both module coherence (0.1) and diversity (0.44).

Visual examination of STAMP's output showed well-defined spatial topics that corresponded to seven key anatomical regions of the hippocampus proper, namely Cornu Ammonis 1 (CA1), CA2, CA3, dentate gyrus (DG), third ventricle (V3), medial habenula (MH) and lateral habenula (LH) (Fig. 2c and Extended Data Fig. 2). In contrast, no other method was able to separate the CA2 and CA3 regions, nor correctly capture the LH region. LDA, NMF and NSFH correctly captured CA1, DG, V3 and MH, but failed to capture the LH and separate CA2 and CA3. LDVAE captured CA1 and DG, but not LH and merged CA2 with CA3, as well as V3 with MH. SpiceMix performed the poorest, failing to capture

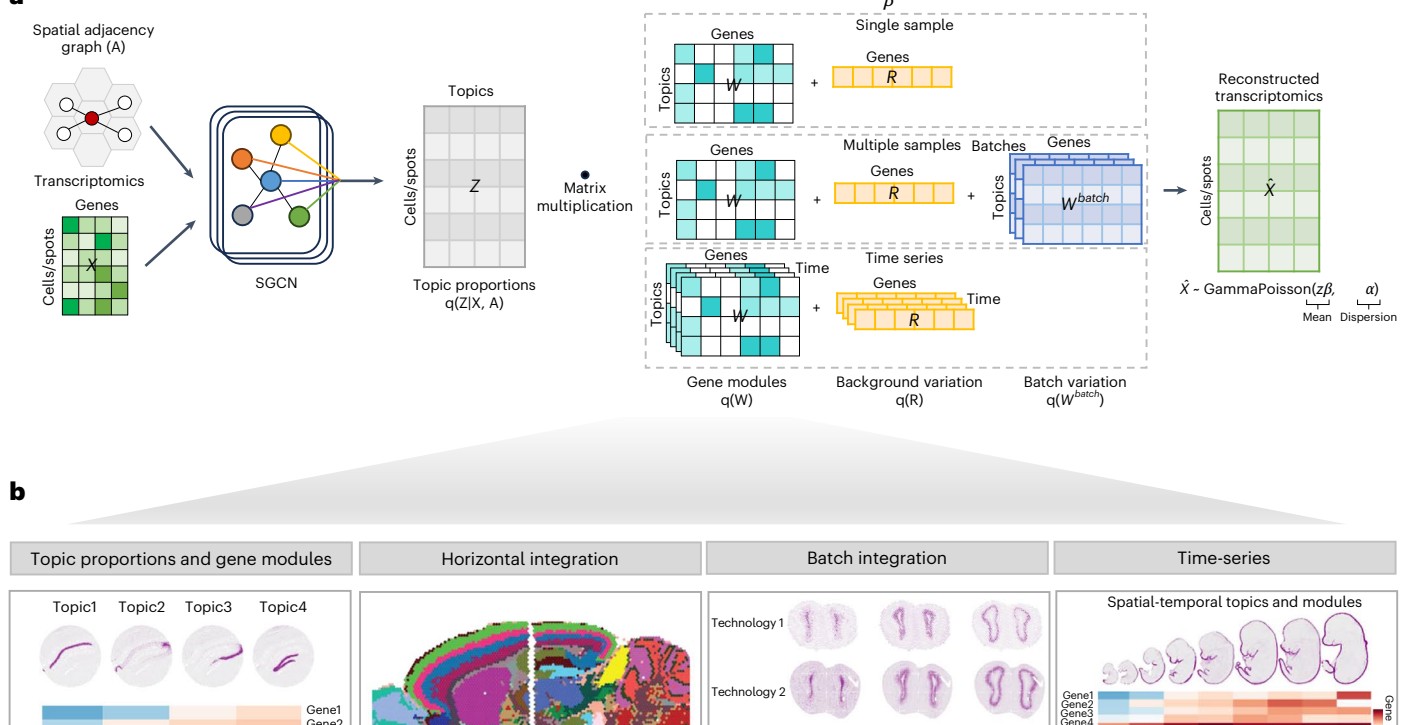

**Fig. 1 | Interpretable deep generative model for spatial transcriptomics analysis. a**, STAMP workflow. STAMP takes in the gene expression values and a neighborhood graph built from the spot locations. STAMP covers three different analysis scenarios, single sample, multi-sample and time-series modules. STAMP uses black-box variational inference and simplified graph convolutions network (SGCN) to find the variational parameters. **b**, Capabilities of STAMP. The gene module scores $w_{kg}$ and topic proportions $z_{nk}$ are outputs of STAMP. These outputs can be used for downstream analysis such as horizontal, vertical integration and time-series analysis.

any anatomical regions (Extended Data Fig. 2 and Supplementary Figs. 1–6). We further investigated the respective gene modules by examining the rankings of published gene markers of the anatomical regions (Fig. 2d). For all regions except CA3, at least one marker was found within the top ten (*Wfs1* and *Pou3f1* in CA1, *Rgs14* in CA2, *C1ql2* and *Prox1* in DG[31,32], *Enpp2*, *Igf2* and *Ttr* in V3, *Tac2* and *Gpr151* in MH and *Cbln1* and *Cbln2* in LH[33,34]). These well-established marker genes for their respective anatomical regions provide evidence that STAMP can generate biologically meaningful topics and gene modules. We then visualized the spatial distributions of STAMP's topic proportions and expressions of the top gene from the respective module, affirming that the topic proportions and gene expression levels are congruent (Fig. 2e).

**STAMP uncovers a topic of cancer-associated fibroblasts**
We next assessed STAMP's ability to uncover cell states and associated gene modules using one sample from a human non-small cell lung cancer (NSCLC) dataset acquired with CosMx SMI[35]. The selected Lung #5-3 sample is composed of 93,206 spots capturing 960 genes (Fig. 3a). Here we set STAMP to return 15 topics and their gene modules. The gene modules were analyzed with DISCO's scEnrichment[36] tool to annotate the topics present (Supplementary Table 1). We found the topic annotations to agree well with the manually annotated cell types in the original study, confirming the accuracy of the topics and their gene modules in capturing cellular states (Extended Data Fig. 3a,b). We next plotted the gene module rankings of known cell type markers[36,37] to verify the gene modules' biological relevance (Fig. 3b). Most cell type markers to be highly ranked within their respective gene modules with at least

one marker in the top 10 (except for tumor edge) and most within the top 20, confirming the gene modules' biological relevance. We also visualized the spatial expression of the top genes per topic and aggregated expression of the top 20 genes. These genes showed high spatial coexpression patterns and accurately correlated with their respective topics, highlighting the coherence between the gene modules and their topics (Supplementary Fig. 7).

STAMP's topics also achieved cell type annotation at a higher resolution than the original study (STAMP in Fig. 3c, original in Extended Data Fig. 3a, comparison in Extended Data Fig. 3b). For example, STAMP segregated the tumor into three domains, tumor (Topic 15), tumor interior (Topic 8) and tumor edge (Topic 9), supported by marker genes. The fibroblasts were also resolved at a higher resolution into two topics of fibroblasts and one of cancer-associated fibroblasts (CAFs). The two fibroblast topics possessed different signatures with higher expression of collagen genes such as *COL3A1* and *COL1A1* in the Topic 6 fibroblasts and proteoglycan genes such as *DCN*, *MMP2* and *LUM* in the Topic 11 fibroblasts, representing different fibroblast subsets (Extended Data Fig. 3c). Collagen proteins make up the bulk of the extracellular matrix (ECM) of connective tissues by forming bundles of fibers that make up the main tissue structure, whereas proteoglycans are soluble glycoproteins that form large complexes with collagen fibers. We also annotated the CAFs (Topic 13) based on reported CAF signature genes including *PDGFRB*, *ACTA2*, *MYH11*, *IGFBP5* and *IGFBP3* (refs. 38–40) (Supplementary Table 1 and Fig. 3b). We found the CAF topic to be concentrated on the exterior of the tumor edge, matching reports of CAFs forming a barrier separating tumor tissues from other stroma regions[41]. We therefore quantitated the spatial relationships between the topics

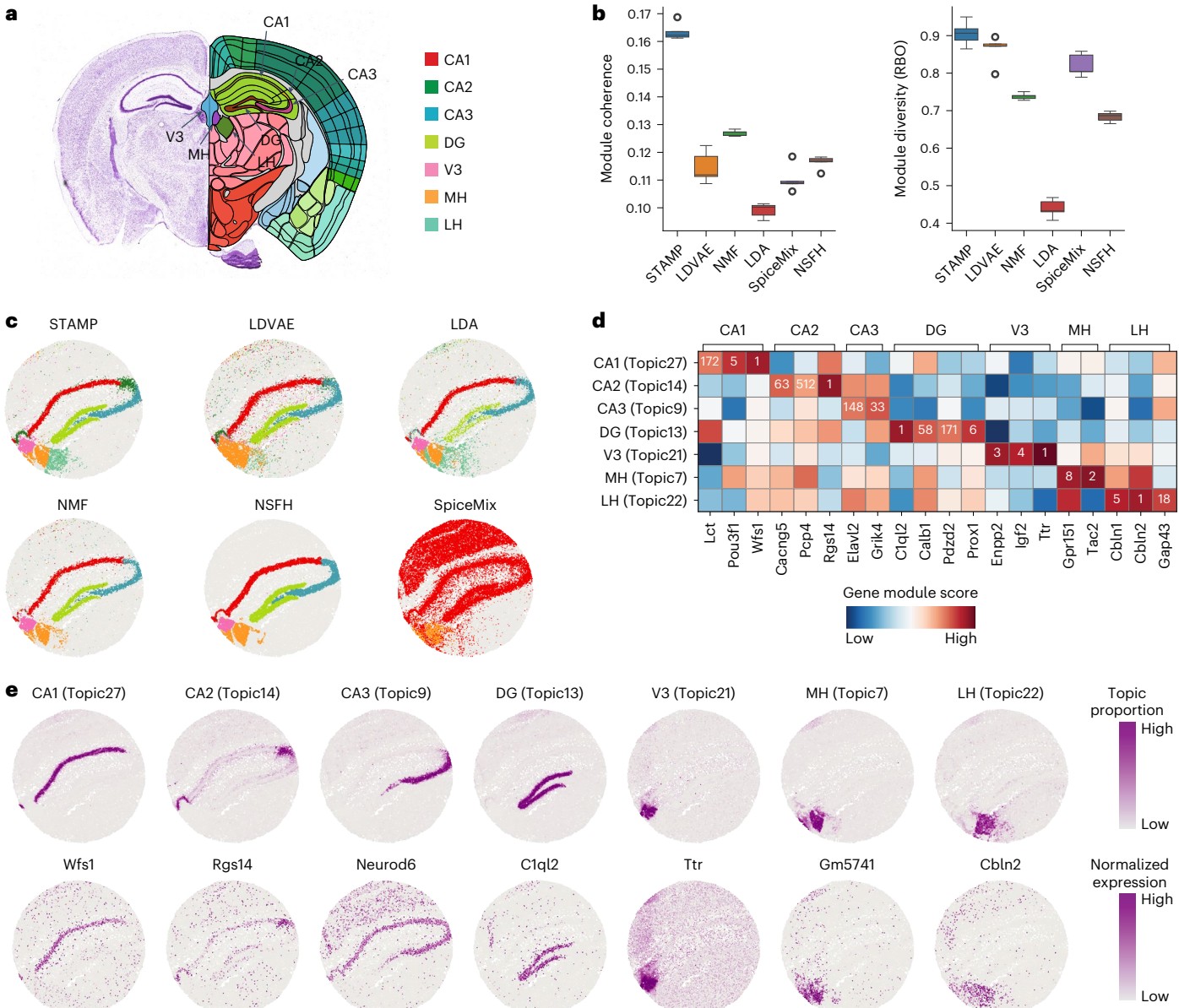

**Fig. 2 | STAMP accurately identifies biologically relevant topics in Slide-seq V2 mouse hippocampus. a**, Allen Mouse Brain Atlas reference with the hippocampus region annotated. **b**, Boxplots of module coherence and module diversity scores of STAMP and the five competing methods obtained over five different runs with different seeds. In the box plot, the center line denotes the median, box limits denote the upper and lower quartiles and whiskers denote 1.5 × interquartile range. **c**, Selected spatial topics related to the hippocampus as captured by the methods. Topics were binarized for ease of visualization. **d**, Gene module rankings (number) and scores (color) of known gene markers for the different hippocampus regions. **e**, Visualization of the spatial topics identified by STAMP and expression levels of the corresponding highest-ranking genes from each gene module.

by computing their co-occurrence probabilities using Squidpy[42]. The computed co-occurrence probability with respect to the CAF topic showed that it has the highest co-occurrence with the tumor edge, thus further strengthening our assigned annotation (Fig. 3d).

As CAFs exert an important influence on the tumor microenvironment to promote inflammation and tumor progression, we were interested in the contributing molecular pathways. We thus characterized the CAF topic by computing its differentially expressed genes with respect to the other fibroblasts (Topics 6 and 11), followed by pathway analysis (Fig. 3e). The analysis revealed upregulated cytokine signaling which can be attributed to inflammatory molecules secreted within the tumor tissue[43]. Fibrosis pathways were also upregulated, suggesting a potential overlap in molecular mechanisms between the CAFs and the development of pulmonary fibrosis pathways[44,45]. Finally, the

wound-healing pathway was also upregulated, which has been reported to contribute to the formation of stroma tissue during epithelial tumor development and is involved in protumor crosstalk with tumor cells[46,47].

## STAMP stitches mouse anterior and posterior brain sections

Due to acquisition technology limitations, separate experiments are needed to capture larger areas of tissue from the same section. To analyze such multi-sample data, we incorporated an additional latent variable into the model to create STAMP with a batch correction capability (Fig. 1a). We first demonstrated STAMP's ability to discover common topics across different samples using two 10x Genomics Visium datasets of mouse brain sagittal sections[48,49], divided into posterior and anterior (Fig. 4a). Here we benchmarked STAMP against NMF, LDA, LDVAE, NSFH and SpiceMix, evaluating them by metrics and visual inspection. We first

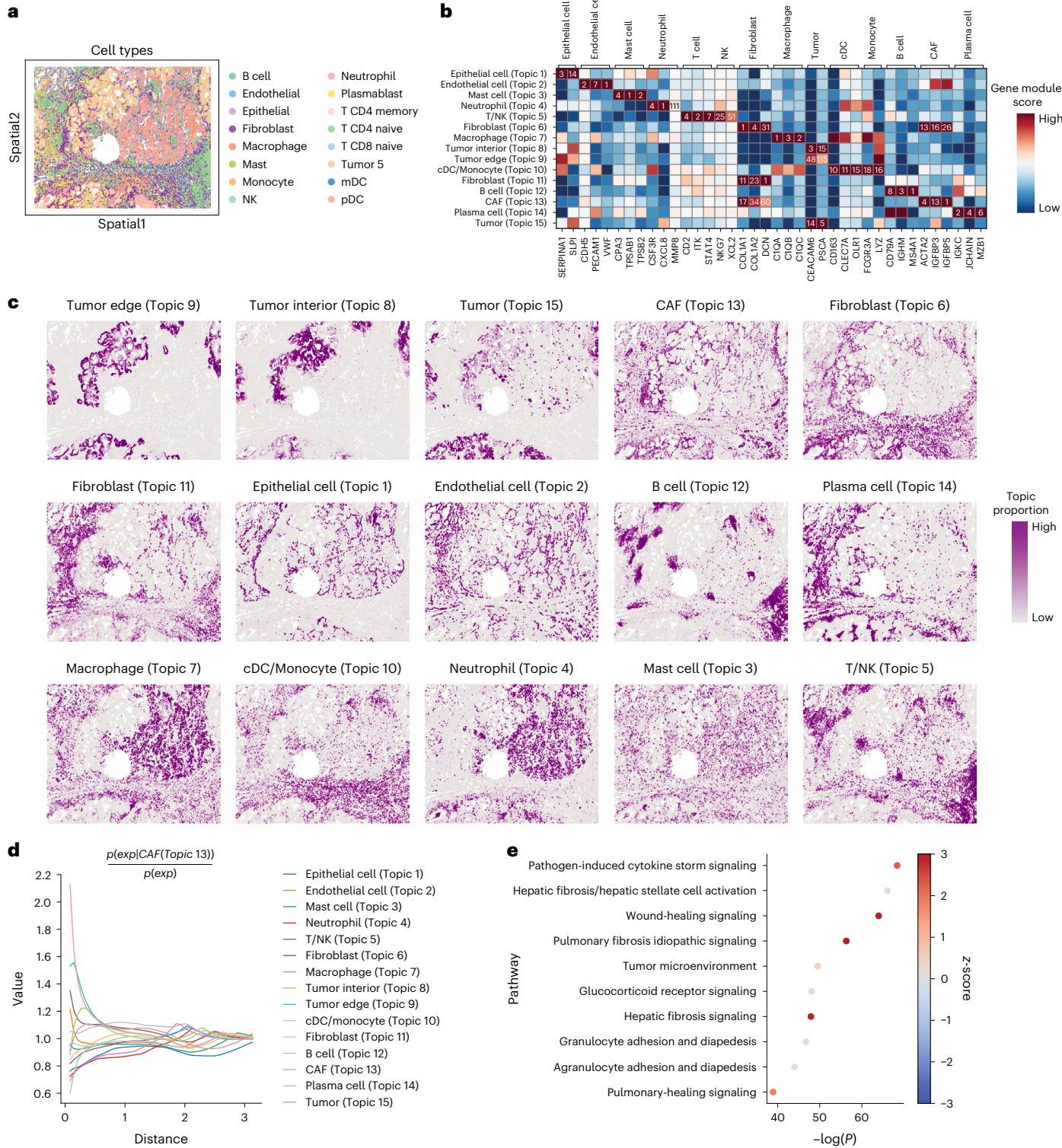

**Fig. 3 | STAMP deconvolutes CAFs from regular fibroblasts in SMI NSCLC data.** **a**, NSCLC sample data acquired using CosMx Spatial Molecular Imager (SMI) with the original cell type annotation. **b**, Gene module rankings (number) and scores (color) of marker genes for each topic. **c**, Spatial plots of topics annotated using their gene modules. **d**, Spatial co-occurrence of different cell types with respect to CAFs (Topic 13) as computed using Squidpy. CAFs are the closest to the tumor edge (Topic 9). **e**, Canonical pathway enrichment analysis obtained from ingenuity pathway analysis (IPA) of CAFs against the other fibroblast populations. Spot color reflects the IPA *z*-score enrichment of CAFs versus fibroblasts, with red indicating predicted pathway activation and blue indicating pathway repression. The *x* axis shows the level of significance via −log$_{10}$ (adj. *P*). Adj. *P* values were calculated with the Fisher's exact test (right tailed) followed by the Benjamini–Hochberg adjustment. NK, natural killer; cDC, conventional dendritic cell; mDC, myeloid dendritic cell; pDC, plasmacytoid dendritic cell.

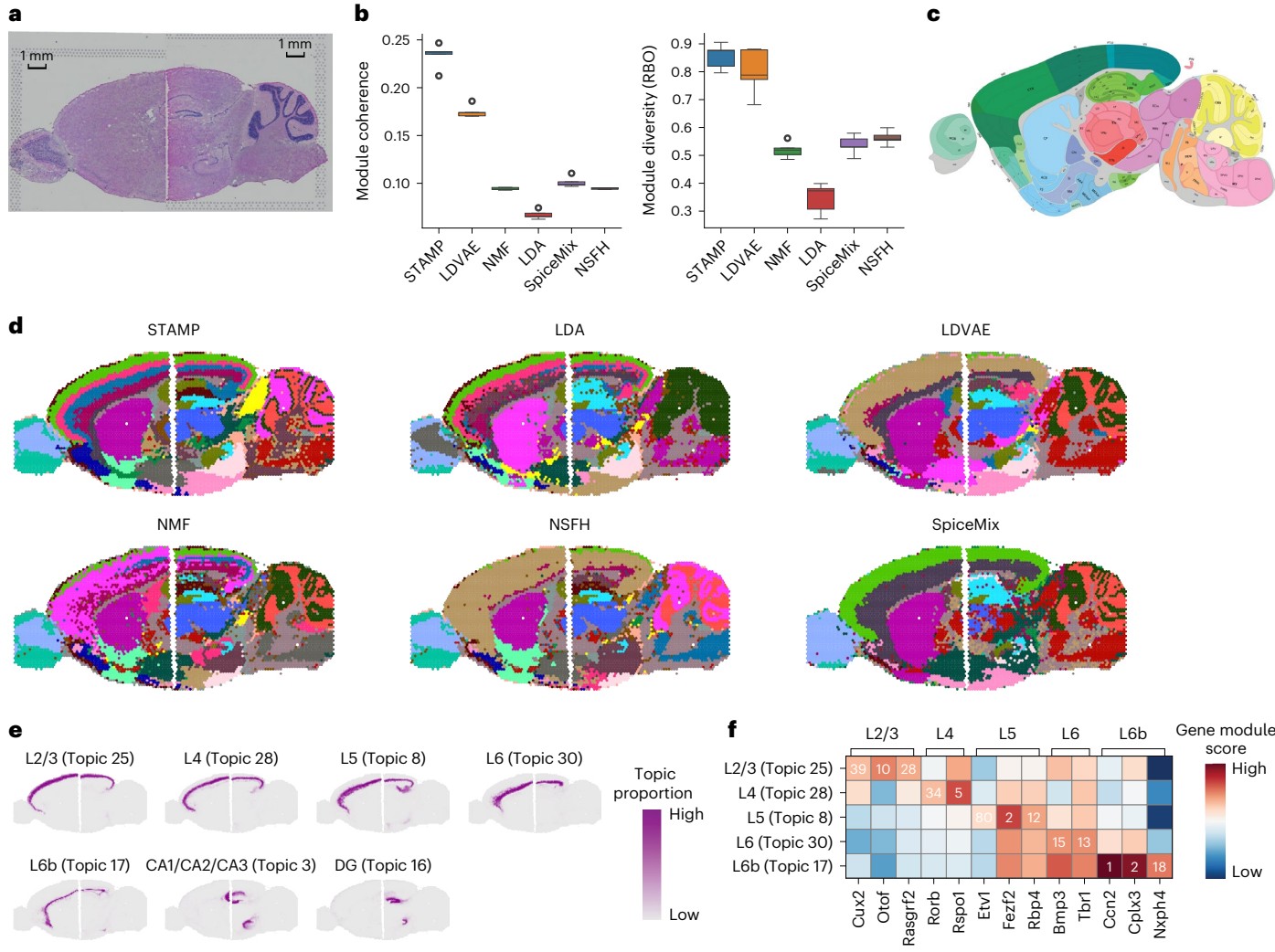

**Fig. 4 | STAMP delineates different cortex layers across multiple Visium mouse brain tissue sections. a**, Hematoxylin and eosin (H&E) image of the mouse brain anterior and posterior sections. **b**, Boxplots of module coherence and module diversity scores of STAMP and five competing methods obtained over five different runs with different seeds. In the box plot, the center line denotes the median, box limits denote the upper and lower quartiles and whiskers denote 1.5 × interquartile range. **c**, Annotated mouse brain sagittal section image from the Allen Mouse Brain Atlas. **d**, Spatial topics returned by STAMP and the five competing methods. Each topic was binarized for ease of visualization in a single figure. **e**, Layers of the cerebral cortex specifically identified by STAMP. **f**, Gene module rankings (number) and scores (color) of known gene markers in different layers of the brain.

quantified their performance in terms of gene module coherence and diversity (Fig. 4b). STAMP exhibited top performance in both module diversity and module coherence. LDVAE was second in both metrics. Like the mouse hippocampus example, LDA attained the worst performance in both metrics. Using the Allen Brain Atlas[11,28] as references (Fig. 4c), we visually compared the outputs of STAMP and competing methods (Fig. 4d and Supplementary Figs. 8–13). Overall, all methods were able to discover topics that were aligned between two sections but not all matched the reference well. STAMP was able to cleanly capture the cerebral cortex layers, the internal structures of the hippocampus (CA and DG) and cerebellum and separated the caudoputamen (CP) and nucleus accumbens (ACB). Among the competing methods, only LDA and NMF were able to separate some of the upper cortex layers, while only NMF and NSFH captured the CA and DG in the hippocampus and separated the CP and ACB. Conversely, the cerebellum's structure was captured by all methods except LDA.

We further investigated STAMP's topics and their associated gene modules, focusing on the cortex layers and hippocampus (Fig. 4e). STAMP was able to identify topics shared by the two brain sections and aligned them well. To verify that the layers were correctly captured, we checked the rankings of reported markers[50–52] in their respective gene modules (Fig. 4f). We found most markers to rank within the top 20 and three layers had one marker ranked within the top 5. This affirmed the ability of STAMP's model in capturing topics and gene modules that are biologically meaningful.

**STAMP identifies shared topics across different technologies**
Here we showcased STAMP's ability to recover topics and gene modules from multiple datasets acquired with different technologies. Here we used three mouse olfactory bulb (Fig. 5a) datasets each acquired with a different technology, Slide-seq V2, Stereo-seq and 10x Genomics Visium, for a total of 148,087 cells. From the Uniform Manifold Approximation and Projection (UMAP) visualization generated with PCA, a substantial batch effect is present in the original data (Fig. 5b). We benchmarked STAMP alongside LDA, LDVAE, NMF, NSFH and SpiceMix to assess data integration performance. We added three more algorithms, DeepST[53], PRECAST and STAligner[54], which provide uninterpretable embeddings but are able to correct for batch effects in spatial data. The methods SpiceMix, DeepST and STAligner were unable to produce results on these datasets due to memory constraints.

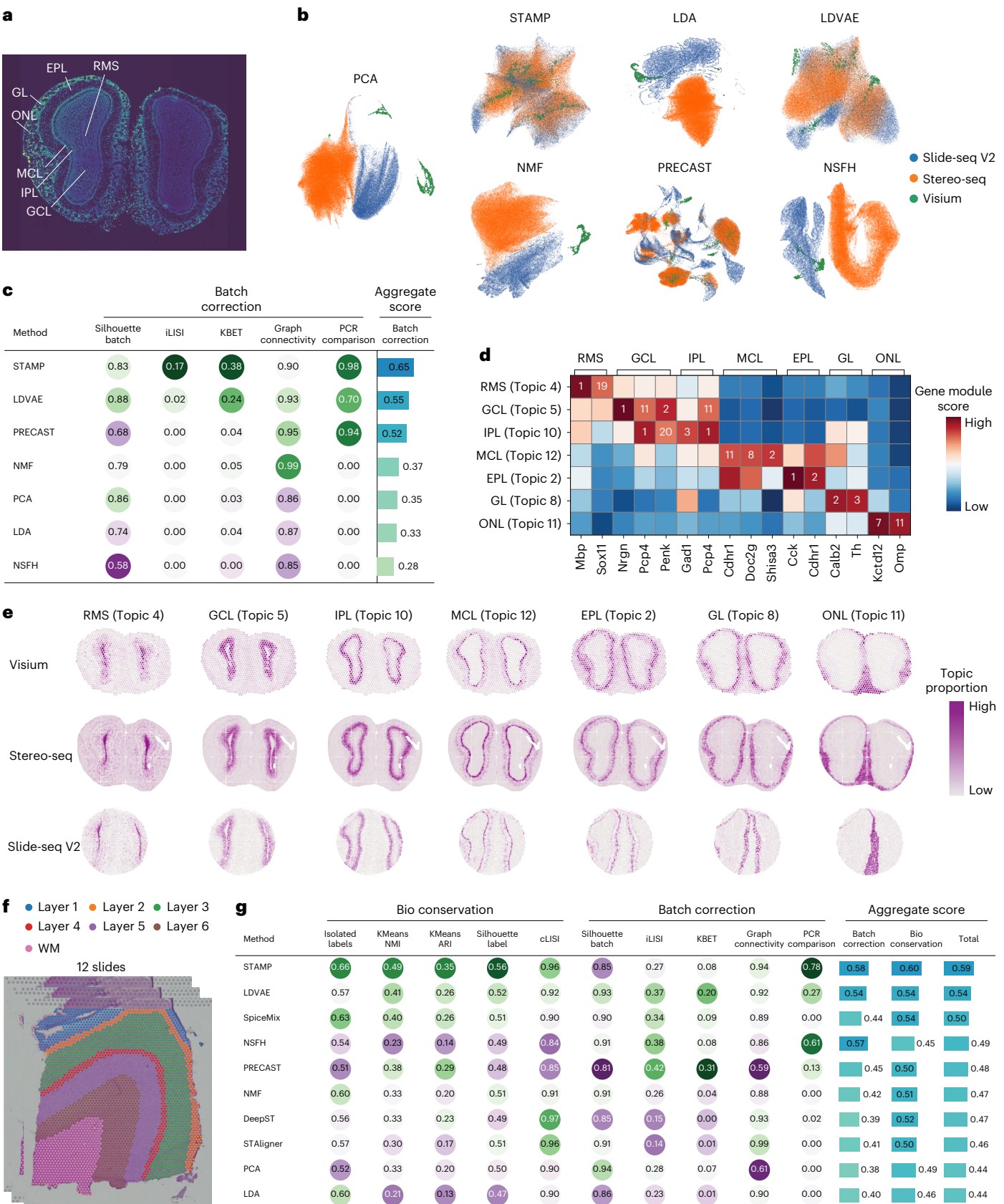

**Fig. 5 | STAMP removes batch effects. a**, Annotated 4,6-diamidino-2-phenylindole (DAPI) stained image illustrating the mouse olfactory bulb's laminar organization. **b**, UMAPs computed using the latent space/topics returned by every method. The UMAPs are colored by their respective technology. **c**, Table displaying the scIB metrics computed for each method. A score of 1 indicates optimal performance. **d**, Gene module rankings (number) and scores (color) of marker genes returned by STAMP for each topic. **e**, Identified topics of different olfactory layers in the 10x Genomics Visium, Stereo-seq and Slide-seq V2 data. **f**, Dataset of 12 DLPFC slices with the original annotation. **g**, Table displaying the scIB metrics and aggregate scores computed for each method. A score of 1 indicates optimal performance.

We fitted 12 topics to the data with all methods and visualized their output with UMAP. In STAMP's output, the three datasets were highly mixed (Fig. 5b). Some mixing could be observed for LDVAE and PRECAST but substantial batch-specific clusters were still visible. For LDA, NMF and NSFH, the datasets remained separated in a batch-specific manner as they are not explicitly designed to handle batch effects. We further quantified the data integration with scIB[55] metrics (silhouette score, iLISI, kBET, graph connectivity and PCR comparison) and the aggregate results (Fig. 5c) broadly matched the UMAP plots. STAMP obtained the best aggregate score while LDVAE was second, followed by PRECAST, NMF, LDA and NSFH ranked at the bottom with aggregate scores comparable to the PCA output, implying poor data integration.

Examining the associated gene modules of STAMP's topics, we found genes associated with the known layers of the mouse olfactory bulb, enabling manual annotation (Fig. 5d and Supplementary Fig. 14). We were able to identify the rostral migratory stream (RMS), granule cell layer (GCL), inner plexiform layer (IPL), mitral cell layer (MCL), external plexiform layer, glomerular layer (GL) and olfactory nerve layer (ONL). Many of the reported markers were highly ranked within their modules, namely *Mbp*[56] (ranked first in RMS), *Nrgn*[57] and *Penk*[58] (ranked first and second, respectively in GCL), *Pcp4* and *Gad1* (ranked first and third respectively in IPL), *Shisa3* and *Doc2g*[59] (ranked second and eighth respectively in MCL), *Calb2* (ref. 60) (ranked second in GL) and *Kctd12* (ref. 56) (ranked seventh in ONL). Of note, the GCL and IPL shared many upregulated genes such as *Pcp4* and *Penk*. We also visualized the topics' spatial distribution in the three datasets (Fig. 5e) and found them to be correctly organized according to the olfactory bulb's structure (Fig. 5a)[61]. This clearly demonstrated STAMP's ability to identify shared topics across data acquired with different technologies and correct for substantial batch effects.

We next ran STAMP on a collection of 12 dorsolateral prefrontal cortex (DLPFC) slices[62] acquired with 10x Genomics Visium (Fig. 5f). The original data was annotated to indicate the cortex layers and white matter. The availability of this ground truth annotation enabled us to test the conservation of biological characteristics (bioconservation) while minimizing batch effects (batch correction) with a comparatively large number of batches. Here we fitted ten topics with STAMP and competing methods. Visualizing the integrated outputs with UMAP, all methods except for PRECAST were able to maintain separation between the cortex layers and white matter (Extended Data Fig. 4a). All the methods mixed the batches to various degrees, but batch-specific regions are highly visible in the DeepST and PRECAST outputs (Extended Data Fig. 4b). For quantitative assessment, we employed five metrics for bioconservation (isolated labels, $k$-means-based NMI, $k$-means-based ARI, silhouette score for annotation labels and cLISI) and five for batch correction (silhouette score for batch, iLISI, kBET, graph connectivity and PCR comparison) from the scIB metrics. In the aggregate scoring, STAMP was top for both bioconservation and batch correction (Fig. 5g). For bioconservation, STAMP was top for all metrics except for cLISI where it trailed DeepST with only a small deficit (0.96 versus 0.97). STAMP was also able to capture the clear layered patterns with its topics (Supplementary Fig. 19).

**STAMP unveils spatiotemporal topics in embryo development**

Last, we demonstrated STAMP's ability to discover spatiotemporally linked topics and associated gene modules from time-series spatial transcriptomic data. We analyzed Stereo-seq data of mouse embryo samples at eight time points from E9.5 to E16.5 for an overall dataset size of more than 540,000 cells[63] (Fig. 6a). To account for dynamic changes in gene expression, the gene modules were allowed to vary across time points with a Gaussian process. This dynamic formulation allows STAMP to capture connected topics across time. We fitted 40 topics that captured matching tissues across different time points, illustrating their developmental trajectories (Extended Data Fig. 5a). The binarized topics (Extended Data Fig. 5b) resolved biological details

that matched or exceeded the dataset's original clustering and annotation (Fig. 6a). It is worth noting that the spatial smoothness of STAMP's topics can be controlled by adjusting the number of layers in the SGCN, where a higher number of layers results in smoother topics. We demonstrated this with the results obtained from the embryo section at E11.5 with different numbers of SGCN layers and used Moran's *I* index to quantitatively demonstrate the changes in spatial smoothness of the topics (Extended Data Fig. 6a,b).

Tissues identified with STAMP's topics include major organs such as the skeleton, liver, heart, skin, jaw and lung. Within the cranial region, STAMP resolved tissues like the choroid plexus and forebrain (Fig. 6b). Examining the individual topics, we could also observe the correspondence of topics across time, tracking tissue development trajectories (Fig. 6b). One example was the heart (Topic 9) that grew in size and our results for E11.5 suggested the formation of heart chambers[64]. Another example is the skin (Topic 21) that started to develop at E9.5 and by E13.5 (ref. 65) we could see a discernible structure enveloping the embryo. For the meninges (Topic 15), it maintained a consistent structural form throughout development. Moreover, STAMP can map the developmental continuum, linking distinct early and late forms that are typically identified as unrelated clusters by conventional clustering analysis. For example, STAMP captured continuous muscle development from dermomyotome to a full-grown muscle in a single topic (Topic 2), whereas the original annotation shows it as two different clusters (Fig. 6c). Notably, the associated gene module contained *Neb*, which is an important skeletal muscle development marker, and its expression had the same spatial distribution as the topic, thus supporting the merging.

The associated gene modules also reflected both the similarities within each topic between time points and the gradual changes across time. In the UMAP visualization, gene modules of the same topic clustered close together (Fig. 6d, left). With PCA, gene modules of the same topic also clustered together. Notably, the principal components also captured the progressive changes of many topics across time (Fig. 6d, right). These ordered and gradual changes highlighted STAMP's ability to model them with the gene modules. They also illustrated the development programs that were shared between tissues and that the gene modules could be further exploited for further biological insights.

To show the biological information captured by the connected gene modules, we focused on two important topics, hematopoiesis (Topic 12) and hepatocyte (Topic 23), which are colocalized in the developing liver (Fig. 6e). During embryonic development, hematopoiesis occurs at the yolk sac during early development before shifting to the liver from seeded hematopoietic stem cells. By E12.5, the liver is the main site of hematopoiesis before a gradual shift to the nearby spleen by E15.5 (ref. 66). Further migration to the bone marrow then begins after E16.5. Consequently, both the hematopoiesis and hepatocyte topics almost entirely overlap spatially from E12.5; however, their gene modules clearly separate them. Using gene set enrichment analysis (GSEA), the hepatocyte topic's gene module showed strong enrichment of lipid processing and other metabolic processes, while gene sets related to hematopoietic process, especially erythrocyte and hemoglobin related metabolic processes, were highly represented for the hematopoiesis topic (Fig. 6f). We further confirmed that the markers of erythrocytes and hepatocytes have high gene module scores (ranked within top 100) within the hematopoiesis[67] and hepatocyte topics[68,69], respectively (Fig. 6g).

## Discussion

In this work, we developed STAMP, a deep probabilistic approach for identifying topics and relevant gene modules in spatial transcriptomics data. To incorporate spatial information, we feed an adjacency matrix built from the spatial locations to STAMP that encodes the spatial adjacency using an SGCN. STAMP also employs a regularized horseshoe prior on the gene modules to encourage structured

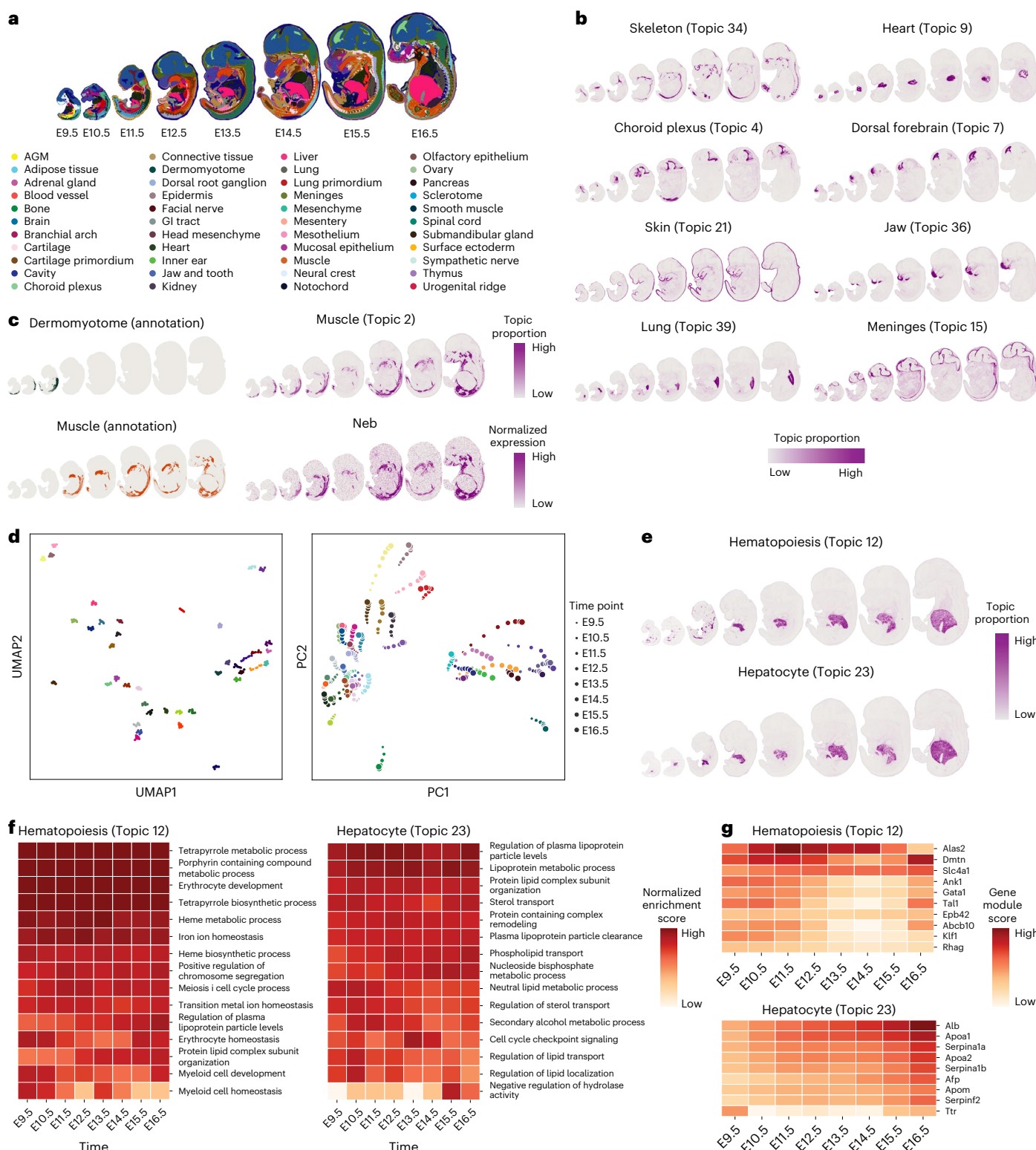

**Fig. 6 | Analysis of time-series Stereo-seq embryo data. a**, Stereo-seq data series of mouse embryo development from E9.5 to E16.5 with the original annotation. **b**, STAMP identified spatial temporal patterns such as the skeleton, heart, skin, jaw, choroid plexus, dorsal forebrain, lung and meninges. All the fitted topics can be found in Extended Data Fig. 5. **c**, Comparison of the original annotation (left) and STAMP's output (Topic 2). STAMP merged the dermomyotome and muscle regions into a single topic. The spatial gene expression of *NEB* (identified from the associated gene module) supports the merger. **d**, UMAP and PCA plots of STAMP's output revealed continuous trajectories of spatially connected gene modules. **e**, Hepatocyte and hematopoiesis topics identified by STAMP. **f**, GSEA normalized enrichment scores computed with the hepatocyte and hematopoiesis topics' gene modules. Top 15 pathways ordered by the normalized enrichment score were selected and shown. **g**, Gene module scores of selected markers in the hepatocyte and hematopoiesis topics. AGM, aorta-gonad-mesonephros; GI, gastrointestinal.

sparsity, which leads to more robust and interpretable estimates. We tested STAMP on several datasets, achieving favorable performance in terms of both module coherence and diversity when compared to other competing methods such as NMF, LDA, LDVAE, NSFH and SpiceMix. In the mouse hippocampus dataset, STAMP identified distinct gene modules associated with different anatomical regions within the hippocampus, capturing the molecular heterogeneity underlying this brain region's functional diversity. With the human lung cancer dataset, STAMP captured the spatially organized gene expression patterns that corresponded to specific tumor regions and cell types. This enabled annotation at a higher resolution than the original study, segregating the tumor into different regions such as tumor edge and tumor interior, as well as resolving additional fibroblast subsets such as CAFs.

We also extended STAMP to integrate multiple datasets and even time-series data. We performed integrative analyses of two mouse brain sagittal sections (anterior and posterior), capturing biologically accurate topics that were also aligned along the shared edge between both sections. STAMP also demonstrated the capability to integrate data acquired from different technologies and batches with data of mouse olfactory bulb sections, as well as human DLPFC sections.

Most notably, we employed STAMP to model spatiotemporally linked topics in a series of developing mouse embryo sections at eight time points. STAMP unveiled intricate anatomical structures at a higher resolution compared to the original study's annotations. By analyzing the associated gene modules in terms of highly ranked genes and enriched pathways, we annotated the higher resolution topics and revealed their biological significance. For example, STAMP accurately captured the liver and hematopoiesis topics with relevant genes highly ranked within their associated gene modules. Notably, the hematopoiesis and liver topics coincided spatially during the time at which the embryonic liver was the main site of hematopoiesis. These results highlight the power of spatial transcriptomics and the utility of STAMP in deciphering complex biological systems.

We also note some of STAMP's limitations and areas for future development. In its current form, STAMP is unable to facilitate comparative analysis between different conditions such as normal versus cancer or mutant versus wild type. Such analyses are important in health and disease-related studies to capture and dissect the detailed phenotypic differences and potentially lead to mechanistic explanations. Therefore, a future development is to modify STAMP to recover topics that are different across conditions. Additionally, a further development is the optional inclusion of previous knowledge as input, such as gene sets or pathways, cell type or niche information, which may be partially available for the tissue of interest. This can help guide the construction of gene modules with prior biological knowledge and potentially achieve better accuracy and biological relevance. Last, another avenue of further development is to extend STAMP to handle spatial multi-omics data[70] or image data, including the mosaic data scenario, where not all datasets have the same data modalities available. The inclusion of additional data modalities increases the information content capture, including orthogonal information, thereby increasing the overall accuracy.

We designed STAMP to be user friendly and capable of processing data from different experimental platforms. STAMP can scale to very large datasets with hundreds of thousands of cells with the largest dataset tested having more than 500,000 cells, ensuring its relevance as the size of datasets grows (Extended Data Fig. 7). All analyses were conducted on a server equipped with an Intel Core i7-8665U CPU and an NVIDIA Titan V GPU with 12 GB of memory.

## Online content

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

## Methods
### STAMP model
**Section 1 model composition.** STAMP is a Bayesian model that decomposes the expression $x_{ng}$ of gene $g$ in cell $n$ into topic proportion $z_{nk}$ (the proportion of the $k$-th topic in cell $n$) and gene embedding $\beta_{kg}$ of gene $g$ in the $k$-th topic. We model the gene expression $x_{ng}$ as a sample drawn from a Gamma Poisson distribution, where $\mu_{ng}$ is the mean expression of gene $g$ in cell $n$ and $\alpha_g$ is the gene-specific dispersion term.

$$x_{ng} \sim \text{GammaPoisson}\left(\mu_{ng}, \alpha_g\right)$$

*Mean term $\mu_{ng}$.* We model the mean term $\mu_{ng}$ as a summation over $k$ topics of the multiplication between the library size $l_n$, the topic proportion $z_{nk}$ and the gene embedding $\beta_{kg}$

$$\mu_{ng} = l_n \sum_k z_{nk} \beta_{kg}$$

where $k$ labels the $k$-th topic.

*Library size $l_n$.* We model the library size as the observed total gene counts for each cell:

$$l_n = \sum_g x_{ng}$$

*Topic proportion $z_{nk}$.* We model the topic proportion as a logistic-normal prior that we softmax over the topic dimension to ensure that the topic proportion per cell sums to 1. We model the covariances across topics in the form of $\mathbf{U}\mathbf{U}^T + \sigma\mathbf{I}$ decomposition, where $\mathbf{I}$ refers to the identity matrix. The rank of the covariance matrix (dimensionality of $u_k$) is a hyperparameter that we set equal to the number of topics.

$$u_k \sim \text{Normal}\left(0, \mathbf{I}\right)$$

$$\sigma \sim \text{HalfCauchy}\left(1\right)$$

$$\tilde{z}_n \sim \text{Normal}\left(0, \mathbf{U}\mathbf{U}^T + \sigma\mathbf{I}\right)$$

$$z_{nk} = \text{softmax}_k\left(\tilde{z}_{nk}\right)$$

*Gene embedding $\beta_{kg}$.* Scenario 1: single sample

We model the gene embedding $\beta_{kg}$ as the sum of the background residual term $r_g$ and the gene module score $w_{kg}$. The residual term $r_g$ captures the gene module shared across all topics, downweighing genes that are highly expressed across all topics. We model it as a normal prior centered around the log of the observed log-normalized mean $\bar{x}_g$, with a small $\epsilon$ set to $1 \times 10^{-8}$ to ensure positivity. The gene module score $w_{kg}$ is modeled with a structured horseshoe prior that introduces shrinkage to the gene module by selectively loading genes onto the topic only if they have sufficient information to overcome the prior, thereby giving rise to more robust gene modules.

$$r_g \sim \text{Normal}\left(\log(\bar{x}_g + \epsilon), 1\right)$$

$$\delta_g \sim \text{HalfCauchy}\left(1\right)$$

$$\tau_k \sim \text{HalfCauchy}\left(1\right)$$

$$\lambda_{kg} \sim \text{HalfCauchy}\left(1\right)$$

$$c \sim \text{InverseGamma}\left(0.5, 0.5\right)$$

$$\tilde{\lambda}_{kg} = \delta_g \tau_k \lambda_{kg}$$

$$w_{kg} \sim \text{Normal}\left(0, \frac{c^2 \tilde{\lambda}_{kg}^2}{c^2 + \tilde{\lambda}_{kg}^2}\right)$$

$$\beta_{kg} = \text{softmax}_g\left(w_{kg} + r_g\right)$$

Each level of the structured horseshoe prior hierarchy contributes to the structured sparsity. Specifically, $\delta_g$ controls the gene-wise sparsity. When $\delta_g$ approaches 0, the prior on $w_{kg}$ approaches a spike at zero for gene $g$, deeming the gene irrelevant for all topics. The same principle applies to $\tau_k$, which encourages topic-wise shrinkage. $\lambda_{kg}$ controls element-wise shrinkage, allowing each gene module to turn off specific genes that are irrelevant to its associated topic. When $c^2 \ll \tilde{\lambda}_{kg}^2$, the prior on $w_{kg}$ approaches normal$(0, c^2)$, therefore regularizing $w_{kg}$ even when the parameters are weakly identified.

Scenario 2: multiple sample

For each batch, we model the gene-wise batch effect of gene $g$ of batch $s$, $\delta_{sg}^{batch}$, as a zero-mean Student's $t$-distribution with a small s.d. of 0.01 and degree of freedom of 10 as we expect most of the genes to be unaffected by batch effects. The Student's $t$-distribution has heavier tails compared to the normal distribution, allowing the possibility of accommodating large batch effect present in the data. The topic-wise batch effect of topic $k$, $\tau_k^{batch}$, is modeled as a β prior, controlling the amount of batch effect present in each topic. The Beta prior puts more mass on both ends (near 0 and 1). The contribution of batch effect $w_{skg}^{batch}$ is therefore given by an outer product of the gene and topic-wise batch effect terms. Last, the batch-specific gene embedding $\beta_{skg}^{batch}$ is the sum of $w_{skg}^{batch}$, gene module $w_{kg}$ and background residual $r_g$. The former two terms are retained from the last scenario.

$$\delta_{sg}^{batch} \sim \text{StudentT}\left(10, 0, 0.01\right)$$

$$\tau_k^{batch} \sim \text{Beta}\left(0.5, 0.5\right)$$

$$w_{skg}^{batch} = \tau_k^{batch} \otimes \delta_{sg}^{batch}$$

$$\beta_{skg}^{batch} = \text{softmax}_g\left(w_{kg} + r_g + w_{skg}^{batch}\right)$$

Scenario 3: time series

In the context of time-series data analysis, we expect the gene module $w_{kg}$ to vary across time points $t$. Here, we model each $w_{kg}$ as an independent Gaussian process with a Matern 3/2 kernel. The Matern 3/2 kernel $\kappa(t, t')$ consists of two hyperparameters, the output variance $\bar{\sigma}_{kg}^2$ that controls the average distance of the function from its mean and the length scale parameter $l$ that determines the smoothness of the function. We fix $l$ to 1 as we want to recover gene modules that are coherent across time and set $\bar{\sigma}_{kg}^2$ to be the regularized horseshoe prior described in the previous section. Last, we put a normal prior on the background residual term $r_{tg}$ at each time point for each gene.

$$\delta_g \sim \text{HalfCauchy}\left(1\right)$$

$$\tau_k \sim \text{HalfCauchy}\left(1\right)$$

$$\lambda_{kg} \sim \text{HalfCauchy}\left(1\right)$$

$$c \sim \text{InverseGamma}\left(0.5, 0.5\right)$$

$$\tilde{\lambda}_{kg} = \delta_g \tau_k \lambda_{kg}$$

$$\bar{\sigma}_{kg}^2 = \frac{c^2 \tilde{\lambda}_{kg}^2}{c^2 + \tilde{\lambda}_{kg}^2}$$

$$\kappa_{kg}(t, t') = \bar{\sigma}_{kg}^2 \left(1 + \frac{\sqrt{3}(|t - t'|)}{l}\right) e^{\left(\frac{\sqrt{3}|t-t'|}{l}\right)}$$

$$w_{kg} \sim GP(0, \mathbf{K}_{kg})$$

$$r_{tg} \sim \text{Normal}(\log(\bar{x}_{tg} + \epsilon), 1)$$

$$\beta_{tkg} = \text{softmax}_g(w_{tkg} + r_{tg})$$

*Dispersion term $\alpha_g$.* We model the square root of the dispersion as a Half-Cauchy prior. The prior places most of its mass toward zero, signaling that most of the genes do not exhibit overdispersion.

$$\sqrt{a_g} \sim \text{Half-Cauchy}(1)$$

**Section 2 inference.** Black-box variational inference is used to approximate the posterior, building on the automatic differentiation variational inference (ADVI) framework from Pyro[71]. The joint probability for the single sample scenario is given by

$$p(\mathbf{X}, \mathbf{\Theta}, \mathbf{Z}) = p(\mathbf{\Delta})p(\mathbf{\Lambda})p(\mathbf{T})p(c)p(\mathbf{W}|\mathbf{T}, \mathbf{\Lambda}, \mathbf{\Delta}, c)p(\mathbf{U})$$
$$p(\sigma)p(\mathbf{Z}|\mathbf{U}, \sigma)p(\mathbf{A})p(\mathbf{R})p(\mathbf{X}|\mathbf{W}, \mathbf{Z}, \mathbf{A}, \mathbf{R}),$$

where $\mathbf{\Theta} = \mathbf{\Delta}, \mathbf{\Lambda}, \mathbf{T}, c, \mathbf{W}, \mathbf{U}, \sigma, \mathbf{A}, \mathbf{R}$ and

$$\mathbf{X} = \{x_{ng}\}, \mathbf{T} = \{\tau_k\}, \Delta = \{\delta_g\}, \Lambda = \{\lambda_{kg}\}, \mathbf{A} = \{\alpha_g\}, \mathbf{W}$$
$$= \{w_{kg}\}, \mathbf{Z} = \{z_{nk}\}, \mathbf{R} = \{r_g\}, \mathbf{U} = \{u_k\}.$$

To approximate the posterior $p(\mathbf{\Theta}|\mathbf{X})$, we choose the mean-field variational family $q_\phi(\mathbf{\Theta}) = \prod q_\phi(\Theta)$, which factorizes into independent distributions for each parameter $\Theta$. Specifically, we utilize normal distributions for the real parameters $\mathbf{W}, \mathbf{U}, \mathbf{R}$ and lognormal distributions for the positive parameters $\mathbf{\Delta}, \mathbf{\Lambda}, \mathbf{T}, c, \sigma$ and $\mathbf{A}$, ensuring that they have the same support as the prior distribution.

To approximate the posterior for $p(\mathbf{Z}|\mathbf{X})$, we make use of simplified graph convolutional network that also takes in spatial information. The simplified graph convolutional network takes in the gene expression counts and the spatial adjacency graph, and outputs the mean $\mathbf{Z}_u$ and variance $\mathbf{Z}_\sigma$ for the variational parameters $q_\phi(\mathbf{Z}|\mathbf{X}, \mathbf{A})$.

$$\tilde{\mathbf{A}} = \mathbf{A} + \mathbf{I}$$

$$\mathbf{S} = \tilde{\mathbf{D}}^{-\frac{1}{2}} \tilde{\mathbf{A}} \tilde{\mathbf{D}}^{-\frac{1}{2}}$$

$$\mathbf{Z}_u, \mathbf{Z}_\sigma = NN([\mathbf{X}, \mathbf{SX}, \dots, \mathbf{S}^l\mathbf{X}])$$

$$q_\phi(\mathbf{Z}|\mathbf{X}, \mathbf{A}) = \text{softmax}_k(\text{Normal}(\mathbf{Z}_u, \mathbf{Z}_\sigma)),$$

where $\mathbf{A}$ is the adjacency matrix, $\mathbf{I}$ is the identity matrix, $\tilde{\mathbf{D}}$ is the degree of $\tilde{\mathbf{A}}$, $\mathbf{X}$ is the gene expression matrix and $NN$ is the neural network used. Therefore, the ELBO is given by

$$L = E_{q_{\phi(\mathbf{\Theta}, \mathbf{Z})}}[\log p(\mathbf{X}, \mathbf{\Theta}, \mathbf{Z}) - \log q_\phi(\mathbf{\Theta}, \mathbf{Z})]$$

where $\phi$ denotes the learnable parameters of both the graph neural network and the variational posteriors. The first term represents the expectation of the joint density with respect to the variational distribution $q$, while the second term is the entropy of the variational distribution. The gradient of the ELBO is given by

$$\nabla L = \nabla_\phi E_{q_{\phi(\mathbf{\Theta}, \mathbf{Z})}}[\log p(\mathbf{X}, \mathbf{\Theta}, \mathbf{Z}) - \log q_\phi(\mathbf{\Theta}, \mathbf{Z})].$$

We compute a noisy, unbiased gradient estimate using unbiased Monte-Carlo estimates that rely on reparametrized gradients[16]. This approach stabilizes the optimization procedure by reducing variance and allows for optimization through stochastic optimization techniques such as Adam[72]. We provided more details of the training and simplified graph convolution network in Supplementary Note 1.

#### Outputs and post-processing

The output topic proportions $z_{nk}$ and gene module scores $w_{kg}$ are taken to be the mean of the variational posteriors $q(z_{nk}|\mathbf{X}, \mathbf{A})$ and $q(w_{kg})$, respectively. We further post-process the gene module scores by downweighing the lowly-expressed genes. The new gene module score is given by

$$w_{kg\_new} = w_{kg} - \log(r_g + \epsilon) + \log(r_g)$$

where $r_g$ is the mean of the variational posterior $q(r_g)$. We set $\epsilon$ to the tenth quantile of $r_g$.

#### Binarization of topics

Each cell was assigned to a topic based on the largest value in its topic proportion.

#### Method comparison

We compared STAMP with several existing methods that can identify topics and their corresponding gene modules. We limited the competing methods to those whose latent topics are non-negative, therefore omitting methods such as PCA and its related extensions such as SpatialPCA and MEFISTO[12,26]. Therefore, we compared STAMP to NMF, linear decoded variational autoencoder (LDVAE), LDA, non-negative spatial matrix factorization hybrid (NSFH) and SpiceMix.

**NMF.** NMF factorizes a given gene expression matrix into two non-negative matrices, which can be interpreted as a set of underlying 'metagenes' and 'metasamples'. We ran NMF from the scikit-learn package with the Kullback divergence loss and set max-iter to 1,000 to avoid convergence errors. The KL loss NMF is equivalent to a NMF with Poisson likelihood.

**LDA.** LDA is a generative statistical model under a family of models named topic models that was originally developed for text count data. It assumes a Dirichlet prior on top of the generating process compared to NMF. We ran LDA from the scikit-learn package with its default parameters.

**LDVAE.** LDVAE is a deep latent factor model which replaces the neural network decoder of scVI[73] with a linear decoder. We ran LDVAE with the default parameters except for the use of a logistic-normal distribution as the latent topics' distributions to enforce positivity.

**NSFH.** Non-negative spatial factorization is a spatially aware probabilistic dimension reduction model based on transformed Gaussian processes and NMF. We ran NSFH with the default parameters, with half of the factors as spatial factors and the other half as nonspatial factors as suggested. We also used 3,000 inducing points.

**SpiceMix.** SpiceMix is a spatially aware probabilistic dimension reduction model that uses NMF and hidden Markov random field to model the spatial dependencies. We ran SpiceMix with the default parameters except for using $k$-means as the initialization scheme to obtain the exact

number of topics as we failed to obtain the right number of topics with the default initialization scheme.

**DeepST.** DeepST is a tool that aligns and integrates multiple spatial transcriptomics datasets. It makes use of data augmentation for pre-processing. It uses graph neural networks to incorporate spatial information and a denoising autoencoder for integration. We ran DeepST with its default parameters.

**GraphST.** GraphST is a tool that aligns and integrates multiple spatial transcriptomics datasets. It makes use graph neural networks with a contrastive loss to incorporate spatial information. We ran GraphST with its default parameters.

**PRECAST.** PRECAST is a probabilistic algorithm that aligns and integrates multiple spatial datasets. Precast combines factor analysis with an intrinsic autoregressive component for integration and incorporating spatial information. We ran PRECAST with the default parameters.

**STAligner.** STAligner is a tool that aligns and integrates multiple spatial transcriptomics datasets. It uses graph attention neural networks to incorporate spatial information and a triplet loss for integration. We ran STAligner with its default parameters.

### Evaluation metrics

To evaluate the gene modules found, we used two metrics popular in topic modeling. The first is module coherence, which we measure with normalized pointwise mutual information (NPMI). NPMI quantifies the co-occurrence of genes, and the module coherence is calculated by taking the mean of NPMI.

$$\text{Module coherence} = \frac{1}{K}\sum_{k=1}^{K}\frac{1}{190}\sum_{i=1}^{20}\sum_{j=i+1}^{20}\frac{\log_2\frac{P(g_i,g_j)}{P(g_i)P(g_j)}}{-\log_2 P(g_i,g_j)}.$$

where $P(g_i, g_j)$ refers to the joint probability of two genes occurring in the same cell, $P(g_i)$ refers to the probability of observing gene $i$ and $K$ is the number of topics. We use the observed counts of each gene to measure the probabilities. To prevent housekeeping or background genes from dominating the metric, we only consider a gene to be present in a cell if its expression falls within the top 25th percentile. We used the top 20 ranked genes from the topic's gene module for our evaluation. The unnormalized version of the metric has been used in single-cell studies[74,75].

The second is module diversity that measures how unique gene modules are using ranked bias overlap (RBO)[30]. RBO compares two gene modules $M_i$ and $M_j$ of equivalent size and returns a number between 0 and 1, where 1 means the two gene modules are identical and 0 means they are completely unique. We calculate RBO scores between each pair of gene modules using the top 20 ranked genes. The module diversity score is then calculated by taking the mean of the minimum (1 − RBO) score per topic. This ensures that modules with a higher diversity obtain a higher score.

$$\text{Module diversity} = \frac{1}{K}\sum_{i=1}^{K}\min\left(1 - \text{RBO}(M_i, M_j) \text{ for } j \in K\right)$$

To evaluate the batch correction capabilities of the different methods, we utilized the scIB metrics. We employed the scib metrics package from https://github.com/YosefLab/scib-metrics to generate the results.

### Data resources and preprocessing

The spatial adjacency graphs were built with the function squidpy. pp.spatial_neighbors with six nearest neighbors (one hexagonal ring) for structured data (10x Genomics Visium) or with the number of neighbors equivalent to 1/1,000 of the whole dataset. Highly variable genes were selected using Scanpy's highly variable genes command with flavor = 'seurat_v3', which follows the Seurat's pipeline.

**Mouse hippocampus data, Slide-seq V2.** We obtained the Slide-seq V2 data from the SeuratData package (https://github.com/satijalab/seurat-data). We then converted it to a h5ad object for further analysis. We filtered out genes that are expressed in less than 1% of the data and cells that have fewer than 100 genes. We then selected for 6,000 highly variable genes.

**Human non-small cell lung cancer data, CosMx SMI.** We obtained the processed Giotto[76] object from https://nanostring.com/products/cosmx-spatial-molecular-imager/nsclc-ffpe-dataset. The data were then transformed into a h5ad object. We filtered out genes that are expressed in less than 1% of the data and cells that have fewer than 50 counts. We then selected for 600 highly variable genes.

**Mouse brain anterior and posterior data, 10x Genomics Visium.** We obtained the 10x Genomics Visium mouse brain data from the 10x Genomics Data Repository (https://www.10xgenomics.com/resources/datasets). The two files are Mouse Brain Serial Section 2 (sagittal–anterior) and Mouse Brain Serial Section 2 (sagittal–posterior). We filtered out genes that are expressed in less than 3% of the cells and mean nonzero expression <1. We then selected for 2,000 highly variable genes.

**DLPFC, 10x Genomics Visium.** The dataset of 12 human DLPFC tissue sections were obtained from https://github.com/LieberInstitute/spatialDLPFC. We filtered out genes that are expressed in less than 3% of the cells. We then selected for 4,000 highly variable genes.

**Mouse olfactory bulb data, Stereo-seq/Slide-seq V2/10x Genomics Visium.** We obtained the Stereo-seq data from https://db.cngb.org/stomics/mosta/download/. The Slide-seq V2 data were obtained from https://singlecell.broadinstitute.org/single_cell/study/SCP815/highly-sensitive-spatial-transcriptomics-at-near-cellular-resolution-with-slide-seqv2#study-download. The 10x Genomics Visium data were obtained from https://www.10xgenomics.com/datasets/adult-mouse-olfactory-bulb-1-standard. We first filtered out cells that have fewer than 50 genes and filtered out genes that are expressed in less than 3%, 1%, 1% of the cells for 10x Genomics Visium, Slide-seq v2 and Stereo-seq, respectively. We concatenated the data and then selected a total of 6,000 highly variable genes.

**Mouse embryo data, Stereo-seq.** We obtained the 50-binned mouse embryo file (Mouse_embryo_all_stage.h5ad) from https://db.cngb.org/stomics/mosta/download/ which included time points from E9.5 to E16.5. We replaced the time point E10.5 with E10.5_E2S1.MOSTA.h5ad. We first filtered out cells that have fewer than 50 genes and genes that are expressed in less than 1% of the cells for each time point. We next selected 6,000 highly variable genes, to which we then applied another filtering of spatially variable genes with Moran's $I$ down to 2,000 genes.

### Over-representation and gene set enrichment analysis of gene modules

We performed over-representation analysis to identify corresponding cell types associated with each topic with the top 20 genes for each topic. We used the enrich function from gseapy[77] to conduct over-representation analysis with our gene modules scores as inputs. For the human CoxMx SMI Lung data, we used the scEnrichment function from the DISCOtoolkit available at https://github.com/JinmiaoChenLab/DISCOtoolkit_py. For the Stereo-seq mouse embryo data, we used the gsea function from gseapy to conduct GSEA with our gene modules score as input. We used the Gene Ontology Biological Process gene sets from MSigDB[78] as our input gene sets.

## Differential gene expression analysis of gene modules

We performed differential expression analysis to identify marker genes of each topic. We used the Gamma Poisson generalized linear models provided by glmGamPoi[79] to fit the coefficients for each topic, followed by a quasi-likelihood ratio test with empirical Bayesian shrinkage to identify differentially expressed genes.

## Computation server employed

All analyses were conducted on a server equipped with an Intel Core i7-8665U CPU and an NVIDIA Titan V GPU with 12 GB of memory.

## Reporting summary

Further information on research design is available in the Nature Portfolio Reporting Summary linked to this article.

## Data availability

The pre-reprocessed data objects used in this study have been uploaded to Zenodo and are freely available at https://doi.org/10.5281/zenodo.8201825 (ref. 80).

## Code availability

An open-source Python implementation using Pyro[71] of STAMP is available at https://github.com/JinmiaoChenLab/scTM. Scripts to reproduce this can be found at Zenodo at https://doi.org/10.5281/zenodo.8201825 (ref. 80).

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

## Acknowledgements

We thank A. M. Vargas Velazquez and Z. Ong for providing interpretation on the mouse embryo and brain data, respectively. We also thank M. Li for providing support on the DISCO scEnrichment analysis. This work was supported by AI, Analytics and Informatics (AI3) Horizontal Technology Programme Office seed grant (Spatial Transcriptomics ST in Conjunction with Graph Neural Networks for Cell–cell Interaction; C211118015) from A*STAR, Singapore; Open Fund Individual Research Grant (Mapping Hematopoietic Lineages of Healthy and High-risk Acute Myeloid Leukemia Patients with FLT3-ITD Mutations using Single-cell Omics; OFIRG18nov-0103) from Ministry of Health, Singapore; Use-Inspired Basic Research Fund (Identify Novel Targets for Cell Type-specific Immunotherapy using Spatial and Single-cell Omics in Conjunction with AI Analytics) from A*STAR, Singapore; the National Research Foundation, Singapore and Singapore Ministry of Health's National Medical Research Council under its Open Fund-Large Collaborative Grant ('OFLCG') (MOH-OFLCG18May-0003).

## Author contributions

J.C. conceptualized and supervised the project. C.Z. designed the model. C.Z. developed the STAMP software. C.Z., K.S.A. and J.C. wrote the manuscript. C.Z. prepared the figures.

## Competing interests

The authors declare no competing interests.

## Additional information

**Extended data** is available for this paper at https://doi.org/10.1038/s41592-024-02463-8.

**Correspondence and requests for materials** should be addressed to Jinmiao Chen.

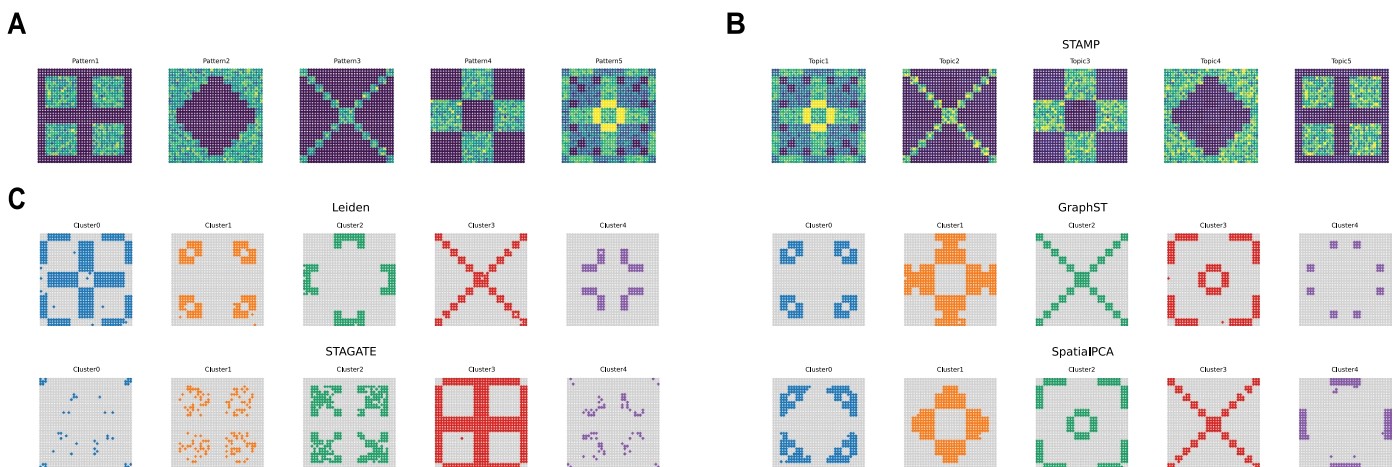

**Extended Data Fig. 1 | STAMP and clustering algorithm results on simulated data. a** Ground truth topics in the simulated data. **b** Topics generated by STAMP. **c** Clusters generated by four different clustering algorithms, Leiden, GraphST, STAGATE, and SpatialPCA.

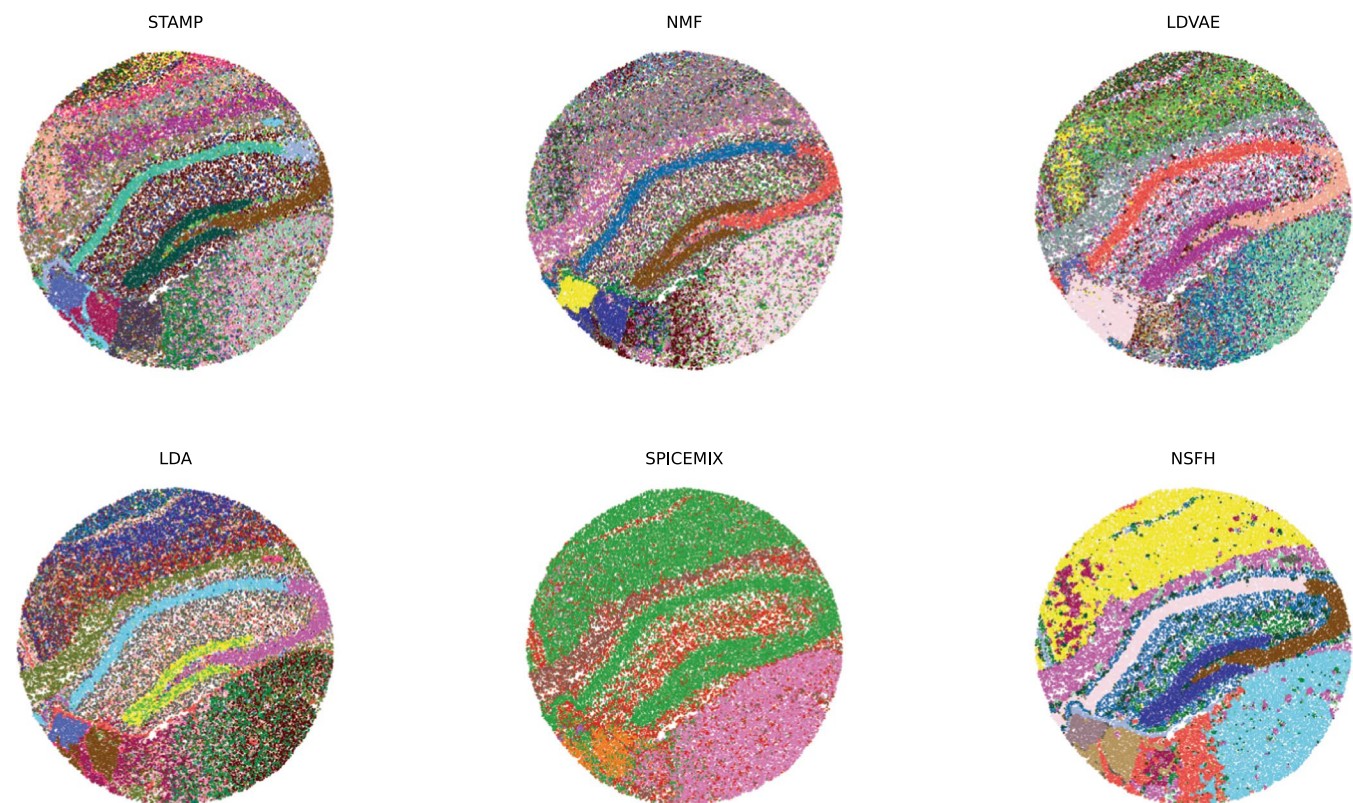

**Extended Data Fig. 2 | Binarized topics of STAMP and other methods.** Binarized topics identified by STAMP, NMF, SpiceMix, LDA, LDVAE and NSFH on the mouse hippocampus data acquired with Slide-seq V2.

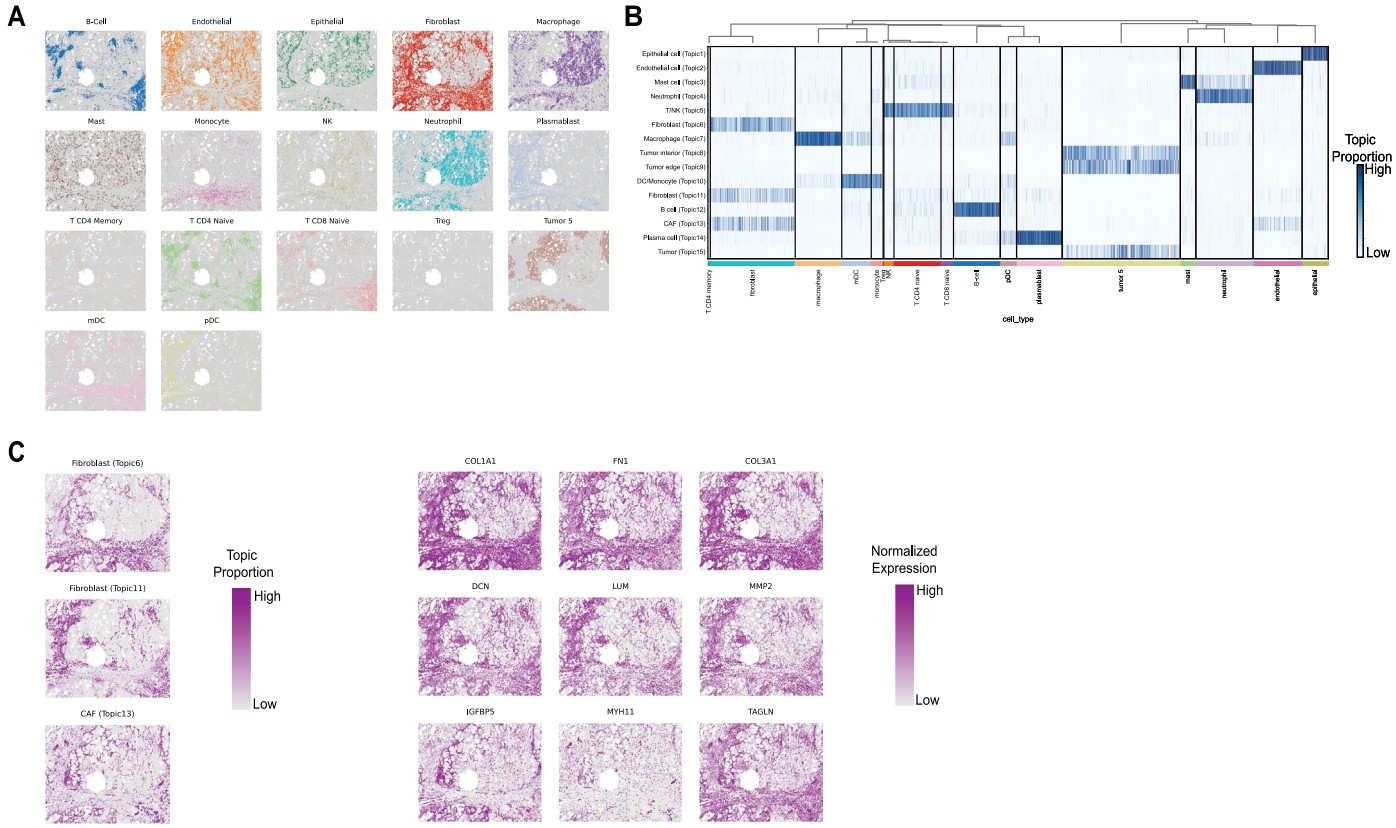

**Extended Data Fig. 3 | Additional results of the SMI NSCLC data. a** Original annotation of the Nanostring SMI NSCLC data. **b** Heatmap of topic proportions returned by STAMP compared against the original annotations. The color intensity denotes topic proportion. **c** Topic proportion of fibroblasts and CAFs (Topics 6 and 11, and 13, respectively), and expression of the top three genes returned in each gene module of the topics.

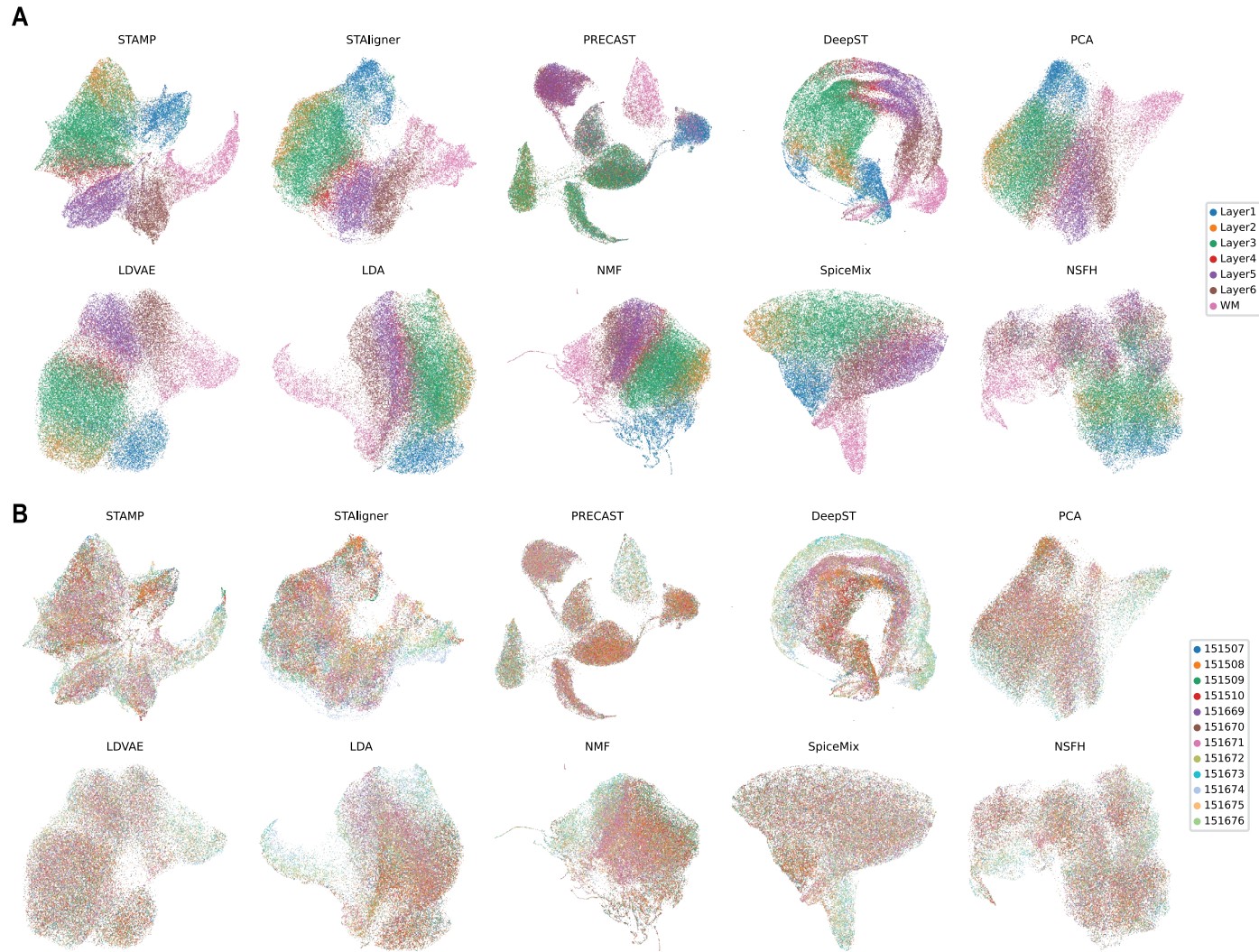

**Extended Data Fig. 4 | Additional results on Visium mouse brain data. a** UMAP of the latent embeddings returned by STAMP and other methods for the 10x Genomics Visium DLPFC dataset, colored by the ground truth annotation. **b** UMAP of the latent embeddings returned by STAMP and other methods for the 10x Genomics Visium DLPFC dataset, colored by library id (sections).

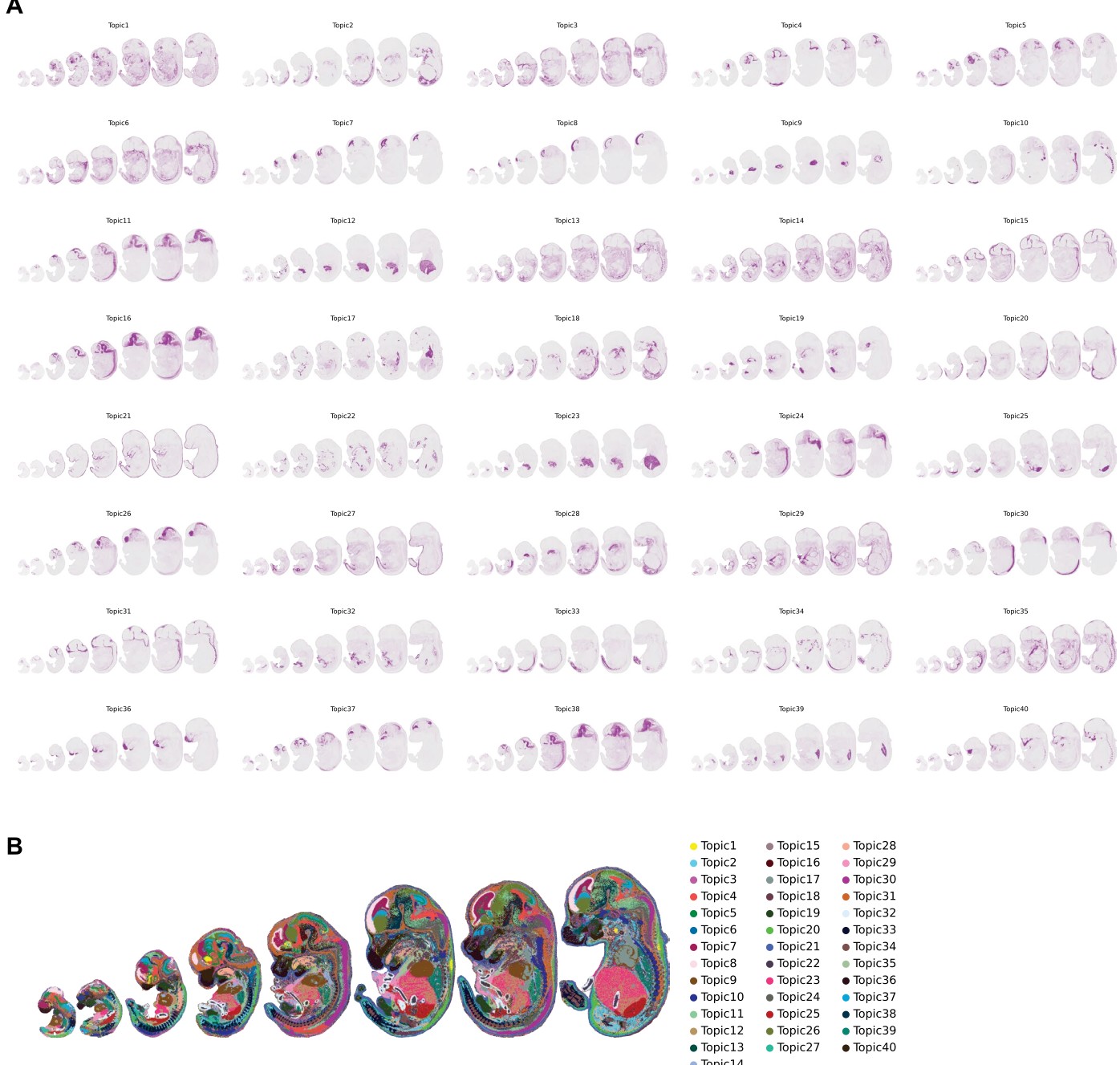

**Extended Data Fig. 5 | Additional results on Stereo-seq mouse embryo data. a** Topic proportions identified by STAMP across all time points of the Stereo-seq mouse embryo data. **b** Binarized topics identified by STAMP across all time points of the Stereo-seq mouse embryo data.

**A**

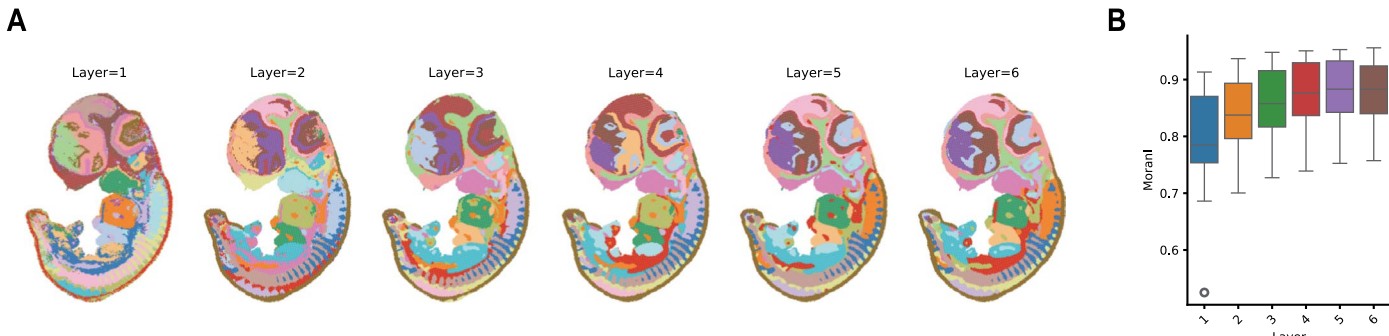

**B**

**Extended Data Fig. 6 | Hyperparameter analysis on STAMP. a** Binarized topics identified by STAMP with different hyperparameter settings on the number of neural network layers. **b** Moran's *I* score of STAMP's topics with different hyperparameter settings on the number of neural network layers.

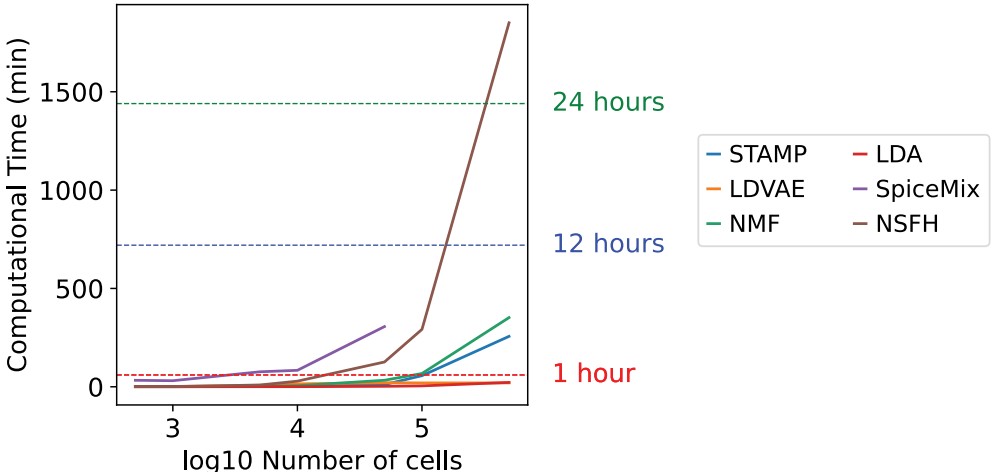

**Extended Data Fig. 7 | Computational time for STAMP and other methods.** Computation time required by STAMP and competing methods for different numbers of input cells. The x-axis is in log scale.

# Reporting Summary

## Statistics

For all statistical analyses, confirm that the following items are present in the figure legend, table legend, main text, or Methods section.

| n/a | Confirmed | |
|---|---|---|
| ☐ | ☒ | The exact sample size (*n*) for each experimental group/condition, given as a discrete number and unit of measurement |
| ☐ | ☒ | A statement on whether measurements were taken from distinct samples or whether the same sample was measured repeatedly |
| ☐ | ☒ | The statistical test(s) used AND whether they are one- or two-sided<br>*Only common tests should be described solely by name; describe more complex techniques in the Methods section.* |
| ☐ | ☒ | A description of all covariates tested |
| ☐ | ☒ | A description of any assumptions or corrections, such as tests of normality and adjustment for multiple comparisons |
| ☐ | ☒ | A full description of the statistical parameters including central tendency (e.g. means) or other basic estimates (e.g. regression coefficient) AND variation (e.g. standard deviation) or associated estimates of uncertainty (e.g. confidence intervals) |
| ☐ | ☒ | For null hypothesis testing, the test statistic (e.g. $F$, $t$, $r$) with confidence intervals, effect sizes, degrees of freedom and $P$ value noted<br>*Give P values as exact values whenever suitable.* |
| ☐ | ☒ | For Bayesian analysis, information on the choice of priors and Markov chain Monte Carlo settings |
| ☒ | ☐ | For hierarchical and complex designs, identification of the appropriate level for tests and full reporting of outcomes |
| ☒ | ☐ | Estimates of effect sizes (e.g. Cohen's *d*, Pearson's *r*), indicating how they were calculated |

*Our web collection on statistics for biologists contains articles on many of the points above.*

## Software and code

Policy information about availability of computer code

| Data collection | No software was used for the data collection |
|---|---|
| Data analysis | scTM v0.1.3(https://github.com/JinmiaoChenLab/scTM), pytorch v2.0.3(https://github.com/pytorch/pytorch), pyro v1.8.4(https://github.com/pyro-ppl/pyro) were used for designing the algorithm.<br>Sklearn v1.2.1(https://github.com/scikit-learn/scikit-learn), Spicemix v1.0.0(https://github.com/ma-compbio/SpiceMix), nsf vN.A. (https://github.com/willtownes/nsf-paper),  scVI v1.0.3 (https://github.com/scverse/scvi-tools) were used for benchmarking.<br>Scanpy v1.9.1(https://github.com/scverse/scanpy), anndata v0.9.1(https://github.com/scverse/anndata), Seurat v4.3.0(https://github.com/satijalab/seurat), glmGamPoi v1.12.2(https://github.com/const-ae/glmGamPoi), squidpy v1.2.2(https://github.com/scverse/squidpy) were used for preprocessing and post-processing of spatial data, STAGATE vN.A. (https://github.com/QIFEIDKN/STAGATE), GraphST vN.A.(https://github.com/JinmiaoChenLab/GraphST), spatialPCA vN.A.(https://github.com/shangll123/SpatialPCA). DeepST v N.A.(https://github.com/JiangBioLab/DeepST) |

For manuscripts utilizing custom algorithms or software that are central to the research but not yet described in published literature, software must be made available to editors and reviewers. We strongly encourage code deposition in a community repository (e.g. GitHub). See the Nature Portfolio guidelines for submitting code & software for further information.

## Data

Policy information about availability of data

All manuscripts must include a data availability statement. This statement should provide the following information, where applicable:

- Accession codes, unique identifiers, or web links for publicly available datasets
- A description of any restrictions on data availability
- For clinical datasets or third party data, please ensure that the statement adheres to our policy

We analyzed a total of 8 spatial transcriptomics datasets for evaluation of latent topics and gene modules. Publicly available data were downloaded from the following websites

1. Mouse hippocampus, Slide-seqV2
The data is downloaded from https://github.com/satijalab/seurat-data.
2. Human non-small cell lung cancer (NSCLC) Data, SMI
The processed Giotto object was downloaded from https://nanostring.com/products/cosmx-spatial-molecular-imager/nsclc-ffpe-dataset.
3. Mouse brain anterior and posterior Data, Visium
The count matrix and spatial data can be downloaded from https://www.10xgenomics.com/resources/datasets. The stitched h5ad can be found in the accompanying zenodo file at https://zenodo.org/records/10988053
4. Mouse olfactory bulb data, Stereo-seq
We obtained the Stereo-seq data from https://db.cngb.org/stomics/mosta/download/. The file to download is Mouse_olfa_S1.h5ad
5.Mouse olfactory bulb data Slide-seq V2
The Slide-seq V2 data was obtained from https://singlecell.broadinstitute.org/single_cell/study/SCP815/highly-sensitive-spatial-transcriptomics-at-near-cellular-resolution-with-slide-seqv2#study-download.
6. Mouse olfactory bulb data 10x Genomics Visium
The 10x Genomics Visium data was obtained from https://www.10xgenomics.com/datasets/adult-mouse-olfactory-bulb-1-standard.
7. Human DLPFC, Visium
The count matrix can be found at https://github.com/LieberInstitute/spatialDLPFC.
8. Mouse Embryo Data E9.5 to E16.5, Stereo-seq
The count matrix and spatial data can be downloaded from https://db.cngb.org/stomics/mosta/download/. The files to download are Mouse_embryo_all_stage.h5ad and E10.5_E1S2.MOSTA.h5ad.
9. MsigDB gene sets
The gene sets can be downloaded from https://www.gsea-msigdb.org/gsea/msigdb

All of the processed data can be found at https://zenodo.org/records/10988053.

## Research involving human participants, their data, or biological material

Policy information about studies with human participants or human data. See also policy information about sex, gender (identity/presentation), and sexual orientation and race, ethnicity and racism.

| | |
|---|---|
| Reporting on sex and gender | N.A. |
| Reporting on race, ethnicity, or other socially relevant groupings | N.A. |
| Population characteristics | N.A. |
| Recruitment | N.A. |
| Ethics oversight | N.A. |

Note that full information on the approval of the study protocol must also be provided in the manuscript.

# Field-specific reporting

Please select the one below that is the best fit for your research. If you are not sure, read the appropriate sections before making your selection.

☒ Life sciences          ☐ Behavioural & social sciences          ☐ Ecological, evolutionary & environmental sciences

For a reference copy of the document with all sections, see nature.com/documents/nr-reporting-summary-flat.pdf

# Life sciences study design

All studies must disclose on these points even when the disclosure is negative.

| | |
|---|---|
| Sample size | We used 8 publicly available data in the manuscript. |

| | |
|---|---|
| Data exclusions | We removed spots and genes by applying the standard preprocessing steps such as removing lowly expressed genes and spots. |
| Replication | N.A. Our experiments did not aim to uncover any mechanistic or intervention effect. Instead, we benchmarked our proposed methodology against competing methods with different datasets, across different technologies |
| Randomization | N.A. Our experiments did not aim to uncover any mechanistic or intervention effect, hence we did not need any controls. |
| Blinding | N.A. Our experiments did not involve human participants and response. |

# Reporting for specific materials, systems and methods

We require information from authors about some types of materials, experimental systems and methods used in many studies. Here, indicate whether each material, system or method listed is relevant to your study. If you are not sure if a list item applies to your research, read the appropriate section before selecting a response.

## Materials & experimental systems

| n/a | Involved in the study |
|---|---|
| ☒ | Antibodies |
| ☒ | Eukaryotic cell lines |
| ☒ | Palaeontology and archaeology |
| ☒ | Animals and other organisms |
| ☒ | Clinical data |
| ☒ | Dual use research of concern |
| ☒ | Plants |

## Methods

| n/a | Involved in the study |
|---|---|
| ☒ | ChIP-seq |
| ☒ | Flow cytometry |
| ☒ | MRI-based neuroimaging |

