## [Peer Review File · Nature Methods]

Interpretable spatially aware dimension reduction of spatial transcriptomics with STAMP

Corresponding Author: Dr Jinmiao Chen

Version 0:

Decision Letter:

21st Sep 2023

Dear Jinmiao,

Your Article, "Interpretable spatially aware dimension reduction of spatial transcriptomics with STAMP", has now been seen by three reviewers. As you will see from their comments below, although the reviewers find your work of considerable potential interest, they have raised a number of concerns. We are interested in the possibility of publishing your paper in Nature Methods, but would like to consider your response to these concerns before we reach a final decision on publication. We therefore invite you to revise your manuscript to address these concerns.

We found the reviews generally constructive, and think that addressing these concerns will strengthen the paper and clarify the benefits of STAMP over existing technologies. We ask that you make a stronger case for improved utility through better clarification of all steps front-to-end and by adding more complete and fair benchmarking. Please also work to ensure that the code/analyses can be reproduced by the referees upon resubmission. Please note, we cannot promise to send the paper back to review until we've seen the revised paper.

Link Redacted

We hope to receive your revised paper within three months. If you cannot send it within this time, please let us know. In this event, we will still be happy to reconsider your paper at a later date so long as nothing similar has been accepted for publication at Nature Methods or published elsewhere.

OPEN SCIENCE REQUIREMENTS

REPORTING SUMMARY AND EDITORIAL POLICY CHECKLISTS

DATA AVAILABILITY

All novel DNA and RNA sequencing data, protein sequences, genetic polymorphisms, linked genotype and phenotype data, gene expression data, macromolecular structures, and proteomics data must be deposited in a publicly accessible database, and accession codes and associated hyperlinks must be provided in the "Data Availability" section.

CODE AVAILABILITY

Please include a "Code Availability" subsection in the Online Methods which details how your custom code is made available. Only in rare cases (where code is not central to the main conclusions of the paper) is the statement "available upon request" allowed (and reasons should be specified).

MATERIALS AVAILABILITY

ORCID

Sincerely,
Rita

Rita Strack, Ph.D.
Senior Editor
Nature Methods

Reviewers' Comments:

Reviewer #1:

Remarks to the Author:

This manuscript by Zhong et al. introduces a new tool, named STAMP, to return low dimensional topics of biologically relevant spatial domains and associated gene modules using spatial transcriptomics (ST) data. STAMP can perform single slice analysis and batch correction for two slices. Furthermore, the interpretability of the STAMP output provides an alternative to differential expression analysis, and the scalability enables large ST data analysis. In general, STAMP appears promising for a range of scenarios they have evaluated it for. I have several major and minor comments that would need to be addressed by the authors.

1. In Figure 2D, marker genes specific to various regions are highlighted, however, they are not top genes highlighted in the corresponding Figure S2 by STAMP. How do these marker genes rank in the STAMP's gene module?

2. In Figure 2H, the authors "plotted the matched top genes from each gene module, which coincided with their expected spatial locations."

How do these matched top genes rank in STAMP's gene module? How can the users pick these matched genes? What is the relationship between these top matched genes with (known) biomarker genes of each topic?

For most Supplementary figures, the authors just highlighted the top 2 genes to demonstrate their high spatial co-expression patterns. These genes accurately correlated with their respective topics. However, for Figure 2H, the authors "plotted the matched top genes from each gene module, which coincided with their expected spatial locations." What threshold would the users need to choose when the top 2 genes by STAMP did not give good results?

3. The authors remark: "We also found commonly used marker genes such as Mbp and Sox11 (rank 5 and 8 respectively) to be highly ranked for the RMS layer, Shisa3 (rank 11) for the MCL layer and Kctd12 (rank 4) for the ONL layer."

The rank of marker genes by STAMP was relatively high, but not the highest. Since identifying biologically relevant topics and gene modules is the essential function of STAMP, the authors need to compare the ranking of marker genes with those of other tools, to demonstrate the strength of their approach.

4. For Figure 2E, the topic proportion is annotated by the spot contribution but how about spots/cells that contribute to more than one topics? Do the authors have a way to determine a threshold for the spot contribution of each topic to allow each spot/cell to only contribute to one topic?

5. It appears that the number of topics is needed to be provided by the user. What is the advantage of topics defined by STAMP relative to the clustering results of other recently published ST clustering tools like SpatialPCA, STAGATE, ADEPT and GraphST and using differential expression analysis to find marker genes? The authors claimed that STAMP possesses interpretability and offers an alternative to differential expression analysis. The authors need to provide more experimental results to demonstrate STAMP's strength in this respect over existing tools.

6. Similar to comment #1, above, the reported cell type markers of the annotated cell types are plotted in Figure 3B, however, they are not top genes highlighted in the corresponding Figure S9. How do these cell type marker genes rank in the STAMP's gene module? The authors need to compare their ranking of marker genes with those of other tools, to demonstrate the strength of their approach.

7. Why separating topic 2 and 8 into two topics is better than merging these two topics?

Depending on the number of predefined topics, if users define a large number of topics, they can generate more topics. The T CD4 memory, T CD4 Naive, and T CD 8 Naive clusters from the annotation of Figure 3A were not identified by any STAMP topics. Additionally, STAMP topic 3 and 9 are both fibroblast. The authors need to better explain the strengths and weaknesses of STAMP in identifying topics.

8. For the batch correction effect in ST data, tools like DeepST, STAligner, PRECAST can learn joint/shared embeddings to perform clustering. The authors need to do a comparison of these tools with their results.

9. It is unclear if the STAMP-BC module can perform batch correction for more than two slices.

10. Similar to comments #1 and #6 above, in Figure 4F, how do these canonical marker genes rank in the STAMP's gene module? In this case too, the authors need to compare the ranks of marker genes with those of other tools, to demonstrate the relative performance of their tool.

11. The pipeline of Figure 1 lacks information and details for STAMP-BC, however, batch correction is part of the main emphasis for the paper. The methods section also needs to provide more details for this part since the input will be more than one ST slice.

12. The equation section in the Methods section omits critical information. Only a few of the variables, such as α_g and d_g , were described adequately. A comprehensive explanation and derivation of the variables and process are necessary to be included in either the main or supplementary methods for a cohesive understanding of STAMP.

13. The explanation of the interpretability of the topics in the article is somewhat unclear. A more robust description of the significance of topic modeling is important to make the study more comprehensible.

Minor:

1. In Figure 3A, please use two or more subfigures to display the 16 gold standard clusters for better visualization. Spatial domains with similar colors are not possible to be distinguished in one Figure by eye. For Figure 3C, the authors need to plot each corresponding golden cluster (Figure 3A) for better comparison and visualization.

2. "The identified topics, gene modules and accompanying scores indicating their contributions are available in the Supplementary (Supplementary Figure S9, Supplementary Table S1)". Something is missing from this statement.

3. The authors need to rename all supplementary Figures and Tables with their actual number. It takes a while to figure which item corresponds to what figure number.

4. On the issue of reproducibility: Example 2 (Figure 4, for joint mouse brain visium): the batch correction effect by STAMP based on the tutorial cannot be reproduced. Please double check the data and code.

5. The three tutorial examples in GitHub do not replicate many results included in the manuscript. It would be highly beneficial if the examples could directly correlate with the results presented within the text, allowing users to better replicate

the findings and understand the outputs by STAMP.

6. Authors highlighted the scalability of STAMP frequently, however, the paper lacked a detailed rationale on the principles behind this scalability.

7. The overall readability of the paper can be improved. Word choice makes some sentences difficult to understand. The manuscript would benefit from another round of editing.

Reviewer #2:

Remarks to the Author:

In this manuscript, Zhong and the co-authors reported a method named STAMP that resolves tissue regions and cell types using spatially-resolved transcriptomic data. Based on a number of different datasets, the authors demonstrated that STAMP performs robustly on different tissue types and recovered associated topics and gene markers. Overall, this manuscript is interesting and has the potential to be useful for the community. However, I have several questions about the technical details of this method as well as some major concerns that weaken my overall enthusiasm.

- STAMP is based on providing concatenated matrix of gene expression, spatially-smooth TX and covariates. (1) When generating the spatial adjacency graph (Figure 1A), I'm curious to know about the process of selecting the number of neighbors for different datasets. Under the method section "Datasets download and preprocessing parameters", it looks like the number of neighbors has been arbitrarily chosen to each dataset. (2) I'm also curious to know about how the number of latent topics is chosen for different datasets. What's the process for determining these parameters?

- In the hippocampus (figure 2) and anterior/posterior brain (figure 4) datasets, the authors benchmarked STAMP with several other methods, including NMF, and demonstrated that STAMP outperforms the others by showing distinct anatomical regions. I have a major concern regarding how the benchmark is performed. It looks like the authors performed NMF simply using gene expression matrix (line 392), whereas the input of STAMP requires a spatially-smooth matrix TX. It is unfortunate that the benchmarking is not quite fair to my views. To add on to this point, in a recently-online method called SPIN (Maher et al, 2023), Supp. Fig. 5b shows an example of NMF on smoothed data. The authors should address this issue and repeat the benchmarking.

- For the two datasets mentioned above, I would also like to know how the authors get the label of tissue region annotation for each topic.

- In line 222-224, the authors found "STAMP-BC resolved the L5 layer into two cell type-specific topics." Any additional reference to validate the two cell types? What are some key marker genes?

- In the mouse embryo development dataset (figure 5), the authors demonstrated "topics that capture the temporal dynamics of embryo development, including the brain and central nervous system (line 248-249)." There are a few similar temporal analysis workflows that use the same data (MOSCOT: Klein et al, 2023; STAligner: Zhou et al, 2022). How is the performance of STAMP compared to these methods? Do they capture similar spatiotemporal patterns?

- The original mouse organogenesis dataset (Chen et al, 2022) contains eight time points. However, it is unfortunate that the authors only showed the results from two of the time points in figure 5. To make the demonstration stronger, the authors should comment on why the two stages are selected and provide the spatiotemporal analysis (figure 5A-D) on more time points.

- The authors "obtained the 50-binned mouse embryo (line 442)" for STAMP. I'm curious to know how the number of bins was chosen. In addition, what's the performance of STAMP on single-cell (or single spot)-resolved, non-binned data?

- Some minor comments:

- "Covariates" were brought up in figure 1A, but no further description/explanation was provided in the manuscript. The authors should provide additional sentences to clarify the term.

- Can the authors provide explanations of the difference between gene topics and modules? It's particularly unclear to me what gene modules mean in the manuscript (line 92 and others).

- To make the paper more focused, I suggest that the authors can reduce or remove the discussions that are not STAMP (i.e. around the other competing methods).

- Texts in some figure panels are very hard to see. For example, in figure 1D and 4B.

Reviewer #3:

Remarks to the Author:

This manuscript applies a deep generative topic model for detecting latent spatial topics and gene modules in spatial transcriptomics data. The method is useful but the manuscript has a lot of clarity issues to be corrected.

1. The graphical model in Figure 1C appears to have some errors. Specifically, c and λ are gene and topic specific. They should not be in the N-rectangle but in the G-rectangle.

2. The preprocessing step to obtain the concatenated gene expression data matrix in Figure 1A is not clearly explained. The steps of building adjacency graph, normalization and smoothing with spatial graph and gene expression to obtain matrix TX should be explained. In addition, the batch effect features in S seems to be binary per experiment?

3. In addition, it is inaccurate to mention STAMP uses graph-convolutional neural network since the spatial information is

only used to create the spatially smoothed gene expression as additional features in TX by an one step random walk on the spatial graph (as shown in Figure 1A). This needs to be clarified.

4. The important measure "Topic proportion" shown in all the results is undefined. How is it calculated? It seems to be the gene column in Beta product with Z?
5. It is not mentioned how the number of topics are decided on each dataset for each method.
6. The details (such as the exact number of spots and genes) of each dataset should be listed in a table in supplementary document.
7. In the definition of STAMP between line 314-321, the L_n term on line 314 is unexplained. It seems to be the diagonal entries of L_n or $L_n L_n^T$ introduced later on line 379? The notation needs to be improved for clarity here.
8. On line 415, how does NPML give measures of topic diversity and topic coherence with one ratio? Are they the nominator and the denominator?

Author Rebuttal letter:

Comments from editor:

We found the reviews generally constructive and think that addressing these concerns will strengthen the paper and clarify the benefits of STAMP over existing technologies. We ask that you make a stronger case for improved utility through better clarification of all steps front-to-end and by adding more complete and fair benchmarking. Please also work to ensure that the code/analyses can be reproduced by the referees upon resubmission. Please note, we cannot promise to send the paper back to review until we've seen the revised paper.

Thank you so much for the time and efforts that you and the reviewers have spent to help us improve STAMP. We have read through the comments carefully and revised STAMP and the manuscript accordingly. The changes made are summarised as follows:

1. The STAMP package has been reworked into three modules for single-sample, multi-sample, and time-series spatial transcriptomics data analysis, respectively.
 - a. Single sample analysis corresponds to the original STAMP algorithm.
 - b. Multi-sample analysis corresponds to the previously named STAMP-BC (batch correction).
 - c. Time-series analysis is a newly added functionality which we have demonstrated in the manuscript with a set of time-series data of mouse embryo development. The associated gene modules are allowed to vary across time points to accommodate gene expression changes.
2. Expansion of testing and benchmarking:
 - a. DeepST, STAligner, and PRECAST have been added to the list of methods tested.
 - b. We tested NMF and similar tools with spatially smoothed data.
 - c. Two examples were added to demonstrate "vertical" data integration, namely mouse olfactory bulb data (acquired with Slide-seq V2, Stereo-seq, and 10x Genomics Visium), and 12 serial dorsolateral prefrontal cortex (DLPFC) sections.
 - d. We added the scIB (single-cell integration benchmark) metrics to assess the vertical integration results.
3. Code revision:
 - a. As described in point 1, the code has been reorganized and expanded in terms of functionality.
 - b. We have added scripts at <https://zenodo.org/records/10988053> for readers to reproduce the results presented in the manuscript.
4. We have revised the manuscript to reflect the above changes to STAMP and also incorporated other suggestions from the reviewers. We have also edited the manuscript to improve the language used and updated the figures with higher quality images and to ensure that the text therein is legible.

Reviewer #1:

Remarks to the Author:

This manuscript by Zhong et al. introduces a new tool, named STAMP, to return low dimensional topics of biologically relevant spatial domains and associated gene modules using spatial transcriptomics (ST) data. STAMP can perform single slice analysis and batch correction for two slices. Furthermore, the interpretability of the STAMP output provides an alternative to differential expression analysis, and the scalability enables large ST data analysis. In general, STAMP appears promising for a range of scenarios they have evaluated it for. I have several major and minor comments that would need to be addressed by the authors.

Response:

We appreciate your effort in reviewing the manuscript and suggesting improvements. We are also happy to hear that you see promise in STAMP. We have significantly revised both the STAMP package and the manuscript to incorporate the suggestions provided and present the new results obtained. We have also significantly upgraded STAMP to enhance its data integration performance when analysing multiple tissue slices. STAMP can now integrate data to discover shared topics in spatially adjacent slices, different slices of the same tissue type (including data acquired with different technologies), and time-series datasets. We have expanded our benchmarking of STAMP to include more methods and added the scIB metrics¹ (Luecken et al 2022) to benchmark data integration. In addition, we appreciate that you have significant concerns on the biological relevance of genes obtained in the topic associated gene modules. We agree that this is of great importance given the goal of interpretability for STAMP as a method. We have sought to address your questions as extensively as possible below. If you have further questions, we are happy to address them.

Comment 1.1: In Figure 2D, marker genes specific to various regions are highlighted, however, they are not top genes highlighted in the corresponding Figure S2 by STAMP. How do these marker genes rank in the STAMP's gene module?

Response 1.1:

Thank you very much for highlighting this. We agree with you that the marker genes shown in Figure 2D are not the top two genes highlighted in the corresponding Figure S2. Here we present how these marker genes rank in the STAMP's gene module and benchmarked against other methods. It should be noted that not all methods were able to reveal the corresponding topics or factors for the seven hippocampus regions depicted in Figure 2D. For example, all methods except STAMP failed to capture CA2 as a specific topic or factor. As such we assigned the closest topic for each region based on the marker gene ranks. Specifically, for the marker genes of each region, we took their average ranks in the different topics and assigned the topic with the highest average rank to the respective region. naïve marker genes' ranks of all tested methods are presented below (Figure R1.1). We also computed the median rank of the markers to summarise the results (last column). Here, STAMP's median marker rank is the best (6) with LDVAE second (14), which are much higher compared to the other methods.

Figure R1.1. Rank of marker genes from various methods applied to the Slide-seq V2 mouse brain data.

Comment 1.2: In Figure 2H, the authors "plotted the matched top genes from each gene module, which coincided with their expected spatial locations." How do these matched top genes rank in STAMP's gene module?

Response 1.2:

Thank you again for these very insightful questions. In the original Figure 2H, we plotted the expression of known marker genes. Here we showed their rankings in STAMP's gene modules (Figure R1.2.1 below and Figure 5D of the revised manuscript). We found that the marker genes all fall within the top 20.

Figure R1.2.1 STAMP's gene module scores and rank of selected marker genes when applied to the Stereo-seq mouse olfactory bulb data.

How can the users pick these matched genes? What is the relationship between these top matched genes with (known) biomarker genes of each topic? For most Supplementary figures, the authors just highlighted the top 2 genes to demonstrate their high spatial co-expression patterns. These genes accurately correlated with their respective topics. However, for Figure 2H, the authors "plotted the matched top genes from each gene module, which coincided with their expected spatial locations." What threshold would the users need to choose when the top 2 genes by STAMP did not give good results?

Users can pick the top matched genes based on the gene module scores. However, we need to highlight that the gene module scores may be inflated in the case of lowly expressed genes. As in the case of DEG analysis, some lowly expressed genes can have large log-values of fold changes, and thus may need to be excluded. Similarly, STAMP can assign gene module scores to lowly expressed genes. In the examples presented in the manuscript, we selected the genes based on both the gene module scores and expression levels. In this revision, we have added a parameter ϵ to downweigh lowly expressed genes. The revised gene module score of gene g in topic k is calculated by:

$$wkg_{new} = \log(rg + \epsilon) - \log(rg) + wkg,$$

where rg is the background expression of gene g . We set ϵ to the 10th quantile of rg . Based on the modified gene module scores, we selected top-ranking genes and investigated how well their spatial

expression patterns correlate with their respective topics.

Here we show the gene ranks obtained and their correlation with the top genes from their modules. We calculated the correlation between topic proportions and gene expression for the top 1000 ranked genes per module to serve as a surrogate measure of coherence (Figure R1.2.2). We observed that the top-ranked genes do present higher correlations, highlighting their high coincidence with their expected spatial locations. Due to space constraints in the Supplementary figures, we plotted only the top two genes to showcase their strong spatial co-expression patterns. Besides the top two genes, other top-ranking genes also accurately correlated with their respective topics, as shown in Figure R1.2.2. To further confirm this, we calculated the average expression of the top 20 genes followed by subtracting the average expression of randomly sampled genes with Scanpy's `score_genes` function. We found that the aggregated expression of these top 20 genes exhibited patterns that closely aligned with STAMP's topic proportions (Figure R1.2.3).

Figure R1.2.2. Correlation of expression of genes with topic proportion. The x-axis denotes the gene ranks in the gene modules and the y-axis denotes the correlation

Figure R1.2.3. Topic proportion, expression of top ranked gene, and aggregated expression of top 20 genes identified from the Stereo-seq mouse olfactory bulb data.

We also note that in the original manuscript, the sentence "We also plotted the matched top genes from each gene module, which coincided with their expected spatial locations." is confusing and misleading. The term "matched" could potentially give the incorrect impression that we selected the genes based on their alignment with their expected spatial locations. More precisely, we plotted reported marker genes that were also highly ranked in the corresponding gene modules, and we found their expression to coincide with their expected spatial locations. In the revised manuscript, we have substantially expanded the analysis and revised the original example presented in Figure 2H and the new results are presented in Figure 5. In the new example, we incorporated two more datasets acquired with the Slide-seq V2 and 10x Genomics Visium technologies, respectively, in addition to the original Stereo-seq data. This was done to assess STAMP's capability to identify coherent topics across data from different technologies. As we have substantially revised the example and therefore the section within the manuscript, the source of confusion has been removed and we have ensured the clarity of the revised text.

Comment 1.3: The authors remark: "We also found commonly used marker genes such as *Mbp* and *Sox11* (rank 5 and 8 respectively) to be highly ranked for the RMS layer, *Shisa3* (rank 11) for the MCL layer and *Kctd12* (rank 4) for the ONL layer." The rank of marker genes by STAMP was relatively high, but not the highest. Since identifying biologically relevant topics and gene modules is the essential function of STAMP, the authors need to compare the ranking of marker genes with those of other tools, to demonstrate the strength of their approach.

Response 1.3:

Thank you for pointing this out. Indeed, the ranks of the marker genes were high but not the highest. Here, we compare STAMP's ranking of marker genes with those of other tools (Figure R1.3). STAMP's median marker rank is the best (2), which are much higher than LDVAE, NSFH, NMF, and LDA. SpiceMix was not included in this comparison as we were unable to successfully run it on this dataset due to memory constraints.

Figure R1.3 Individual rank of marker genes in gene modules of different methods for the Stereo-seq mouse olfactory bulb dataset.

In this revision, we have moved the results of this example to Figure 5, where we ran STAMP on three mouse olfactory bulb datasets each acquired with three different technologies. As three datasets were jointly analysed in the revised example, the resulting gene modules obtained were slightly different from the earlier one generated with a single dataset. Specifically, *Mbp* and *Sox11* now rank 1st and 19th respectively for the RMS layer, *Shisa3* ranks 2nd for the MCL layer, and *Kctd12* ranks 7th for the ONL layer.

We fully agree that identifying biologically relevant topics and gene modules is the essential function of STAMP. Nevertheless, we also would like to comment that spatial transcriptomics combines spatial location and transcriptomics measurements, thereby offering opportunities to identify new marker genes that were not reported previously but correlate well with the observed structures. Moreover, many of the reported markers are proteins, especially surface proteins, while STAMP analyses the transcriptome.

Comment 1.4: For Figure 2E, the topic proportion is annotated by the spot contribution but how about spots/cells that contribute to more than one topic? Do the authors have a way to determine a

threshold for the spot contribution of each topic to allow each spot/cell to only contribute to one topic.

Response 1.4:

Thank you for raising this relevant question. In STAMP, a spot/cell is often associated to multiple topics with varying proportions with the sum of proportions across topics equals to one per spot/cell. In some scenarios such as the mouse brain data (Figure 2E), the highlighted spots/cells could be unambiguously assigned to one dominant topic that scored a much higher proportion than the remaining topics. In other scenarios like the mouse embryo data (Figure 6F in the revised manuscript), the foetal liver was assigned to two major topics representing haematopoiesis and hepatocyte properties, respectively. For this case, we were able to interpret the results based on our prior knowledge of the foetal liver being the major organ that supports both active erythro-myeloid haematopoiesis and hepatic progenitor cell proliferation and differentiation into hepatocytes. The latter case exemplifies how STAMP's soft clustering approach coupled with the gene modules enables us to identify multiple phenotypes that can be present in specific spots/cells. Nevertheless, we agree that for less well studied or highly heterogeneous tissues such as tumours, spots/cells with no dominant topic and/or gene modules with no clear phenotype can pose significant issues in identification and assignment.

At present, we do not have a method to determine a threshold for the spot contribution of each topic to allow each spot/cell to be assigned to one topic. When applicable based on the user's needs, a spot/cell can be assigned to one single topic that shows the highest topic proportion. Alternatively, the topic proportions are a set of latent embeddings and can be used as input for clustering analysis. Both approaches can ensure that each spot/cell only contributes to a single topic or cluster.

Comment 1.5: It appears that the number of topics is needed to be provided by the user. What is the advantage of topics defined by STAMP relative to the clustering results of other recently published ST clustering tools like SpatialPCA, STAGATE, ADEPT, and GraphST and using differential expression analysis to find marker genes? The authors claimed that STAMP possesses interpretability and offers an alternative to differential expression analysis. The authors need to provide more experimental results to demonstrate STAMP's strength in this respect over existing tools.

Response 1.5:

By allowing users to select the number of topics, users can explore the data by experimenting with different numbers of topics based on the tissue complexity and desired analysis granularity. We do not believe this to be a disadvantage. We also offer the Evidence Lower Bound (ELBO) metric as a guiding criterion to users. In our experience, we found it to be effective. Here, we showcase this by presenting an example using the 10x Genomics Visium mouse brain dataset to illustrate our selection of the optimal number of topics. Since the ELBO is stochastic, we utilized the locally weighted scatterplot smoothing (loess) to smooth out the computed ELBO (Figure R1.5.1). Here we noticed that the ELBO converged at around 30 topics.

Metrics such as ELBO are useful in guiding the clustering process, especially at the first round of analysis. However, other considerations such as desired analysis granularity, analysis goals, relevant biological knowledge of the tissue under study, and even data quality can strongly influence the appropriate number of topics for each analysis. Therefore, we believe that the choice of number of topics should be determined by the user's needs.

Figure R1.5.1 ELBO plot of number of topics selected for the 10x Genomics Visium mouse brain dataset.

Although both STAMP and clustering analysis require the users to specify the number of topics or clusters, there are fundamental differences. One key difference between clustering and topic modelling is that the typical clustering approach assigns a spot/cell to only one cluster, while topic modelling gives a probabilistic topic composition to each spot/cell. The latter is a form of soft clustering, which allows each spot/cell to possess multiple identities (i.e. topics). Another major difference between clusters and topics is that each topic is associated with a gene module wherein the genes are ranked by their contribution to the respective topic. The topic associated gene modules are an alternative to the DEGs computed after hard clustering and they can be used to derive biological interpretations for each topic through gene set or pathway enrichment analysis. Consequently, each spot/cell can be associated with multiple biological interpretations (e.g. multiple sets of enriched pathways) with varying proportions. In contrast, a hard clustering followed by differential gene expression analysis results in each spot/cell being assigned to one cluster and associated with one set of DEGs, leading to one biological interpretation that is shared by all cells within the same cluster. Consequently, we cannot uncover phenotypes exhibited by cellular subsets

within each cluster, nor easily link them between such subsets across hard clusters.

To demonstrate the differences and advantages of STAMP over clustering analysis, we used a modified 'quilt' simulator2 to generate simulated spatial data with known ground truth factors and tested STAMP, Leiden clustering, SpatialPCA, GraphST, and STAGATE (Supplementary Figures S28-S30 in Supplementary Note 1 of the revised manuscript, and Figure R1.5.2 below). This dataset is made of five different spatial patterns that are overlapping in their spatial locations. Hence, one spot can be associated with more than one pattern. As a result, clustering methods such as Leiden were not able to recover the ground truth factors as they could only assign each spot to a single cluster. However, topic models like STAMP were able to recover the five overlapping ground truth factors. We also tested another category of methods that are neither topic modelling nor matrix factorization, namely STAGATE, GraphST, and SpatialPCA, which also generate dimension reduced representation from spatial transcriptomics data. Here, we employed them to derive a low-dimensional representation of the simulated data, followed by clustering analysis using the k-means algorithm. Again, the clustering analysis assigned each spot to a single cluster and thus were also unable to identify the overlapping spatial patterns.

Ground Truth Factors

STAMP

Leiden

SpatialPCA + k-means

GraphST + k-means
STAGATE + k-means

Figure R1.5.2 Comparing STAMP with clustering methods on simulation data with overlapping factors.

Lastly, STAMP and clustering analysis are not mutually exclusive. As a dimension reduction method, STAMP produces low dimensional topic proportions that can also be used in clustering and other downstream analyses.

Comment 1.6: Similar to comment #1, above, the reported cell type markers of the annotated cell types are plotted in Figure 3B, however, they are not top genes highlighted in the corresponding Figure S9. How do these cell type marker genes rank in the STAMP's gene module? The authors need to compare their ranking of marker genes with those of other tools, to demonstrate the strength of their approach.

Response 1.6:

Thank you for the comment and suggestion. We extracted the rankings of reported cell type markers for the different tools and presented them below (Figure R1.6). No results are available for Spicemix as we were unable to finish running Spicemix on this dataset due to memory issues. As Spicemix requires storing the full cell-cell graph in the GPU memory during training, this dataset required more memory than the 12GB GPU memory available on our server. Here we find that STAMP slightly outperforms the other methods in the ranking of the marker genes.

Figure R1.6 Ranking of known marker genes by different methods for the SMI NSCLC data.

Comment 1.7: Why separating topic 2 and 8 into two topics is better than merging these two topics? Depending on the number of predefined topics, if users define a large number of topics, they can generate more topics. The T CD4 memory, T CD4 Naive, and T CD 8 Naive clusters from the annotation of Figure 3A were not identified by any STAMP topics. Additionally, STAMP topic 3 and 9 are both fibroblasts. The authors need to better explain the strengths and weaknesses of STAMP in identifying topics.

Response 1.7:

In our initial manuscript, Topic 2 and Topic 8 represented the tumour edge and CAFs, respectively. The tumour edge is a heterogenous region that can contain cancer cells, fibroblasts, immune cells, healthy epithelial cells, and others, which can be further dissected. STAMP divided this region into two major topics, the tumour edge topic associated with cancer markers CEACAM6 and PSCA, and the CAF topic with CAF markers (revised Figure 3B), indicating they were topics with distinctly different gene modules. Using spatial co-occurrence analysis, we found the CAF topic to concentrate on the exterior of the tumour edge (revised Figure 3D), matching reports of CAFs forming a barrier separating tumour tissues from the other stromal regions.

In the revised manuscript, Topics 2 and 8 in the original example now correspond to Topics 9 and 13, respectively. Again, we observed that Topics 9 and 13 showed distinct gene modules despite their co-localization in the same region. Topic 9 that corresponded to the tumour edge had tumour markers highly ranked, while the CAF topic's (Topic 13) gene module was more similar to the fibroblasts' (Figure R1.7.1).

Figure R1.7.1 Gene module scores of marker genes returned by STAMP for each topic. Number indicates the rank and colour denotes the gene module score.

STAMP allows users to choose the number of topics and thus higher granularity is achievable with a higher number of topics selected. Indeed, at 15 topics, the T CD4 memory, T CD4 Naïve, and T CD 8 Naïve cells from the original annotation shown in Figure 3A were not identified by any STAMP topics. When we increased the number of topics to 20 (Figure R1.7.2), we uncovered T/NK sub-populations, namely Topic 6 associated with CD4 T cell markers (IL7R, CD2) and Topic 13 associated with CD8 T or NK cell markers (CCL5, KLRK1) (Figure R1.7.3). These two topics were previously combined when STAMP was set to run for 15 topics.

We also compared STAMP's topic proportions against the original annotation (Figure R1.7.5) and found that Topic 6 aligned with the annotated CD4 T cells and Topic 13 with the annotated CD8 T and NK cells, confirming the cell type identities of Topics 6 and 13. This example shows that with increased number of topics, STAMP was able to reveal the subsets of T and NK cells. Notably, STAMP also revealed two additional topics (Topics 4 and 15) relating to plasmablast subsets (Figure R1.7.4) that were absent in the original annotation. The top genes associated with Topic 4 pertain to the IgG heavy chain while highly ranked genes of Topic 15 include the IgA heavy chain, suggesting the presence of different plasmablast subtypes.

Figure R1.7.2 Topic proportions generated by STAMP for 20 topics on the SMI NSCLC data.

Figure R1.7.3 Topic proportions and expression of top two genes for Topics 6 and 13 returned by STAMP for 20 topics.

Figure R1.7.4 Topic proportions of Topics 4 and 15 and expression of their respective top two genes returned by STAMP for 20 topics.

Figure R1.7.5 Comparison of STAMP's topics against the original annotations. Colour represents topic proportion.

Based on the comments and suggestions, we have revised the STAMP model and the analyses generated. As a result, the previous fibroblast Topics 3 and 9 now correspond to Topic 6 and 11, respectively (Figure R1.7.6). Although the cells in both topics were fibroblasts, the two topics had distinct gene modules and molecular characteristics, which warranted the division into different subtypes. The top genes in the gene module of Topic 6 were COL3A1, COL1A1, and FN1, genes encoding collagen macromolecules that are the main structural protein of animal tissues. The top genes in Topic 11's gene module were DCN, MMP2, and LUM, which belong to the proteoglycan

family. Proteoglycans are also key components of the extracellular matrix where they form complexes with the collagen fibres. We have elaborated further on these differences in the main text (Result section: STAMP uncovers a topic of cancer associated fibroblasts and associated gene module in lung cancer).

Figure R1.7.6 Topic proportions of Topics 6 and 11 and expression of their respective top two genes returned by STAMP for 15 topics.

Lastly, we would like to discuss the strengths and weaknesses of STAMP in identifying topics. As mentioned earlier, users need to specify the number of topics when running STAMP. With different numbers of topics, users can study tissue complexity at different granularity. Similarly, other methods such as LDVAE, NMF, LDA, NSFH, and SpiceMix also require users to specify the number of factors or topics. Compared to these methods, STAMP produces topics that are more accurate as demonstrated in our revised manuscript. Furthermore, STAMP also performs better than other methods in the batch effect correction task (revised Figure 5). Most importantly, STAMP is currently the only method that can be applied to time series spatial transcriptomics data and output time point specific gene modules (revised Figure 6). Lastly, we would like to point out one of the caveats of STAMP. The current version of STAMP does not allow users to zoom into a topic for sub-topic modelling. Users can only increase the number of topics and apply that to all spots/cells. With clustering algorithms, users can cluster spots/cells in a hierarchical manner whereby they can select one specific cluster to perform further sub-clustering. In the future, we plan to address this limitation by upgrading STAMP to version 2.0 that will allow users to perform hierarchical topic modelling with STAMP.

Comment 1.8: For the batch correction effect in ST data, tools like DeepST, STAligner, and PRECAST can learn joint/shared embeddings to perform clustering. The authors need to do a comparison of these tools with their results.

Response 1.8:

Thank you for this insightful comment. It is true that for the batch correction task, tools like DeepST, STAligner, and PRECAST can learn embeddings shared by different data batches. In the revised manuscript, we compared STAMP with DeepST, STAligner, and PRECAST using two examples (revised Figure 5 and Figure R1.8 below). The first example comprises mouse olfactory bulb datasets acquired using three different technology platforms, namely Slide-seq, Stereo-seq, and 10x Genomics Visium. Prior to batch correction, noticeable technical variations were present, and the clustering of cells predominantly reflected the distinct technology platforms (Figure 5B, PCA plot). We ran STAMP and PRECAST batch correction with the sequencing technology set as the batch factor. STAMP was able to evenly mix the cells from different technologies (revised Figure 5B). On the other hand, PRECAST's output showed less uniform mixing for the technology batches with noticeable separation in certain areas. Such visual inspection was supported by the quantitative scIB metrics¹ for assessing data integration (revised Figure 5C). Due to a time constraint of 24 hours or memory constraint (12GB of GPU memory available on our server), we failed to successfully run DeepST and STAligner on this example.

In the second example, we benchmarked STAMP against DeepST, STAligner, and PRECAST on the dataset of 12 DLPFC slices captured with 10x Genomics Visium (revised Figure 5F-G). This dataset is accompanied by manual annotations in the original study, which provides a ground truth for evaluating the conservation of biological variation. We employed STAMP and competing methods to integrate the data, using the different slices as batch factor. Again, we found that STAMP performed favourably compared to the other methods both visually and quantitatively in terms of the scIB metrics for biological conservation and batch correction. This example also demonstrated that STAMP can be used to integrate as many as 12 batches. We plan to test its capabilities beyond 12 samples once we gain access to a larger dataset.

Figure R1.8. Batch effect correction of the mouse olfactory bulb dataset (A-E) and the DLPFC 10x Genomics Visium dataset (F, G).

Comment 1.9: It is unclear if the STAMP-BC module can perform batch correction for more than two slices.

Response 1.9:

Thank you for raising this important point. We agree that the initial manuscript did not make clear if STAMP can perform batch correction for more than two slices. Yes, STAMP can integrate more than two slices. We have added the results of two examples in Figure 5 of the revised manuscript (also in Figure R1.8 shown above) to demonstrate STAMP's ability to handle batch effects across more than

two slices. The first example consisted of data from three mouse olfactory bulb sections each acquired with different technologies while the second was composed of 12 serial sections acquired with 10x Genomics Visium. Both examples have been presented in the previous Response 1.8 above.

Comment 1.10: Similar to comments #1 and #6 above, in Figure 4F, how do these canonical marker genes rank in the STAMP's gene module? In this case too, the authors need to compare the ranks of marker genes with those of other tools, to demonstrate the relative performance of their tool.

Response 1.10:

Here we present the rankings of canonical markers in the output of the different tools. We also updated our list of canonical markers with additional ISH (in-situ hybridization) derived markers to increase the coverage. Overall, the average rank of markers in STAMP's output was the best with most markers within the top 20. In this example, NSFH and STAMP scored the joint top by the median rank.

Figure R1.10 Rank of marker genes returned by different methods on the 10x Genomics Visium mouse brain data.

Comment 1.11: The pipeline of Figure 1 lacks information and details for STAMP-BC, however, batch correction is part of the main emphasis for the paper. The methods section also needs to provide more details for this part since the input will be more than one ST slice.

Response 1.11:

We agree that our description of STAMP and its related figure can be improved. We deeply apologise for the lack of details. At present, STAMP has been reworked into three modules for single-sample, multi-sample, and time-series spatial transcriptomics data analysis.

1. Single sample analysis corresponds to the original STAMP algorithm.
2. Multi-sample analysis corresponds to the previously named STAMP-BC (batch correction).
3. Time-series analysis is a newly added functionality which we demonstrated with a set of time-series data of mouse embryo development. Associated gene modules are allowed to vary across time points to accommodate gene expression changes.

Alongside the algorithmic changes, we have extensively revised the Methods section and Figure 1, as well as added Supplementary Note 2 to better describe the different variants of STAMP.

Comment 1.12: The equation section in the Methods section omits critical information. Only a few of the variables, such as α_g and d_g , were described adequately. A comprehensive explanation and derivation of the variables and process are necessary to be included in either the main or supplementary methods for a cohesive understanding of STAMP.

Response 1.12:

Again, we agree that the Methods section lacks critical information. The manuscript has been greatly revised to provide much more details on STAMP's model in the Methods section and Supplementary Note 2.

Comment 1.13: The explanation of the interpretability of the topics in the article is somewhat unclear. A more robust description of the significance of topic modelling is important to make the study more comprehensible.

Response 1.13:

We apologize for the lack of clarity on this key aspect. Here, we refer to interpretability as the user being able to directly understand the contribution of each input towards the output. Within the context of STAMP, the input is the gene expression with its spatial context and the output are the topics and associated gene modules. Here, the gene modules provide interpretability through its genes and their rankings within. The higher the rank, the greater the gene contributes to the topic being modelled.

Minor:

Comment 1.14.1: In Figure 3A, please use two or more subfigures to display the 16 gold standard clusters for better visualization. Spatial domains with similar colours are not possible to be distinguished in one Figure by eye. For Figure 3C, the authors need to plot each corresponding golden cluster (Figure 3A) for better comparison and visualization.

Response 1.14.1: Thank you for these very useful and practical suggestions. We agree with you that in Figure 3A, using two or more subfigures to display the 16 gold standard clusters will provide better visualization. In the revised manuscript, we have added the subfigures as Supplementary Figure S9 of the revised manuscript (also shown below in Figure R1.14.1). We also agree that plotting each corresponding golden standard cluster can provide better comparison and visualization for Figure 3C. However, we would like to highlight that the original annotation was automatically generated with the software from Nanostring, as described in the original publication, and hence might not be of quality comparable to manual annotation.

Figure R1.14.1 Original annotation of the SMI NSCLC data.

Comment 1.14.2: "The identified topics, gene modules and accompanying scores indicating their contributions are available in the Supplementary (Supplementary Figure S9, Supplementary Table S1)". Something is missing from this statement.

Response 1.14.2:

Thank you for spotting this error. We have revised this sentence to "The identified topics, gene modules and accompanying scores indicating their contributions are available in the Supplementary Figure S10".

Comment 1.14.3: The authors need to rename all supplementary Figures and Tables with their actual number. It takes a while to figure which item corresponds to what figure number.

Response 1.14.3:

We apologise for the labelling issues among the figures and tables. We have now placed the supplementary figures/tables alongside their actual numbers and legends in a single Supplementary File for ease of reference.

Comment 1.14.4: On the issue of reproducibility: Example 2 (Figure 4, for joint mouse brain Visium): the batch correction effect by STAMP based on the tutorial cannot be reproduced. Please double check the data and code.

Response 1.14.4:

We apologize for the lack of reproducibility with our examples presented and the code provided. Initially we uploaded the wrong dataset for the second example, and that led to the issue of reproducibility. In this revision, we have significantly improved the STAMP model and updated our manuscript and code accordingly. We have also checked and verified that the provided code and example datasets will accurately reproduce the output presented in the manuscript.

Comment 1.14.5: The three tutorial examples in GitHub do not replicate many results included in the manuscript. It would be highly beneficial if the examples could directly correlate with the results presented within the text, allowing users to better replicate the findings and understand the outputs by STAMP.

Response 1.14.5:

To enable users to reproduce the results presented in the manuscript, we have added the relevant scripts at <https://zenodo.org/records/10988053>. Users can run them to replicate our results. We have also added more tutorial examples using the data employed in the manuscript at <https://github.com/JinmiaoChenLab/scTM/>.

Comment 1.14.6: Authors highlighted the scalability of STAMP frequently; however, the paper lacked a detailed rationale on the principles behind this scalability.

Response 1.14.6:

The scalability of STAMP primarily arises from the use of simplified graph convolutions that enables mini batching for concurrent computation. This is the key reason for STAMP's scalability with modern multi-core computing resources such as GPU computing. In contrast, most other neural network algorithms that employ graph convolutional or attention networks, such as GraphST, DeepST, and STAGATE, are unable to scale to large datasets without extremely high memory usage. This is due to the need to fit the whole graph in memory during training. When large amounts of GPU memory is unavailable, switching to CPU-based computation will significantly increase the runtime. Graph

subsampling techniques such as GraphSage³ and GraphSaint⁴ can alleviate memory requirements but at the cost of introducing bias. Thus, such approaches are often not used in current spatial algorithms. NSFH, a method that does not use graphs or neural networks, is also not scalable as it uses Gaussian processes which scales cubically with the number of datapoints/inducing points. Here we compared the runtime scaling of STAMP and competing methods using the Stereo-seq mouse embryo dataset (Figure R1.14.6). We found that STAMP was slower than LDA and LDVAE which do not use spatial information. However, STAMP's speed was comparable to NMF's, a matrix factorization method that also does not employ spatial information. In contrast, NSFH and SpiceMix scaled poorly with the number of cells. For the case of 100k cells, SpiceMix encountered memory limitations on our server equipped with a GPU with 12GB memory, as it had to load the entire 100k x 100k cell-cell adjacency matrix into memory during training. This prevents SpiceMix from analysing large datasets without exorbitant memory requirements.

Figure R1.14.6 Runtime scaling of STAMP and other methods on which Stereo-seq mouse embryo data.

Comment 1.14.7: The overall readability of the paper can be improved. Word choice makes some sentences difficult to understand. The manuscript would benefit from another round of editing.

Response 1.14.7: Thank you very much for your review of the manuscript and your questions and suggestions. We agree that there was significant room for improvement, and we have worked towards this aim. We have carefully edited the manuscript, especially the Method section, to improve clarity and readability. If you spot further issues, please do not hesitate to point them out.

Reviewer #2:

Remarks to the Author:

In this manuscript, Zhong and the co-authors reported a method named STAMP that resolves tissue regions and cell types using spatially resolved transcriptomic data. Based on a number of different datasets, the authors demonstrated that STAMP performs robustly on different tissue types and recovered associated topics and gene markers. Overall, this manuscript is interesting and has the potential to be useful for the community. However, I have several questions about the technical details of this method as well as some major concerns that weaken my overall enthusiasm.

We thank you for the effort spent reviewing our work. We have extensively revised the STAMP package and updated the manuscript accordingly. We have also enhanced our figures to ensure better image quality and text legibility. Notably, we have expanded STAMP to enable analysis of time-series spatial data. We have used the full mouse embryo development dataset of eight time points to demonstrate STAMP's capability in this regard. STAMP was able to capture related topics with biological relevance across different time points. We have also revised our benchmarking with more methods and metrics. Finally, we have also sought to address your other concerns in the responses below, including the use of smoothed data as input for NMF and the choice of parameters. If you have further concerns, we will be glad to investigate them.

Comment 2.1: STAMP is based on providing concatenated matrix of gene expression, spatially smooth TX and covariates. (1) When generating the spatial adjacency graph (Figure 1A), I'm curious to know about the process of selecting the number of neighbours for different datasets. Under the method section *Datasets download and pre-processing parameters*, it looks like the number of neighbours has been arbitrarily chosen to each dataset. (2) I'm also curious to know about how the number of latent topics is chosen for different datasets. What's the process for determining these parameters?

Response 2.1:

Thank you for these insightful questions. We apologise for the lack of clarity on the choice of parameters used in STAMP. Here we provide details as follows to elaborate on how to select an optimal value or range of values for the number of neighbours and number of latent topics.

As the number of neighbours can have a large impact on STAMP's overall performance, we performed a sensitivity analysis on the number of neighbours and presented the results in both the Supplementary Note 3 and below. Here we employed the mouse brain dataset acquired with 10x Genomics Visium for sensitivity analysis. As the 10x Genomics Visium spots are arranged in a hexagonal grid, we explored different numbers of rings around a central spot to define its neighbourhood. The first ring (at a distance of 1) comprises six spots that are immediate neighbours, while the second ring (at a distance of 2) encompasses the 12 spots surrounding the first ring, and this pattern continues for subsequent rings. To quantify the impact of the number of neighbours, we employed three different metrics: the previously defined ELBO (the lower, the better), module

coherence (the higher, the better), and module diversity (the higher, the better). As the number of neighbours increased, we found that ELBO increased, module coherence fluctuated with no clear trend, and module diversity decreased. Taking all three metrics into consideration, we opted for a single ring (with 6 neighbours) for the 10x Genomics Visium acquired data used the examples. The optimal number of neighbours was selected in a similar manner for other examples. Figure R2.1.1 Sensitive analysis on the number of neighbours.

We next examined the selection of the number of topics. As the number of topics is a user selected parameter, users can explore the data by experimenting with different topic numbers. Data quality and tissue complexity are factors that influence the choice on topic number. To aid selection, we offer the Evidence Lower Bound (ELBO) metric as a guiding criterion for the selection, which we found to be effective. Here, we illustrate it by presenting an example using the 10x Genomics Visium mouse brain dataset to showcase our selection of a suitable topic number. Since the ELBO is stochastic, we utilized locally weighted scatterplot smoothing (loess) to smooth out the ELBO. We noticed that the ELBO converged at around 30 topics for the dataset, leading us to choose 30 as the number of topics. However, we do believe that, similar to clustering, selecting the number of topics is also dependent on the desired level of granularity. Increasing the number of topics enhances the granularity of topics and likewise increases the resolution cell types or structures uncovered, as we have demonstrated in our response to Comment 1.7.

Figure R2.1.2 Plot of ELBO with respect to the number of topics for the 10x Genomics Visium mouse brain data.

Comment 2.2: In the hippocampus (figure 2) and anterior/posterior brain (figure 4) datasets, the authors benchmarked STAMP with several other methods, including NMF, and demonstrated that STAMP outperforms the others by showing distinct anatomical regions. I have a major concern regarding how the benchmark is performed. It looks like the authors performed NMF simply using gene expression matrix (line 392), whereas the input of STAMP requires a spatially smooth matrix TX. It is unfortunate that the benchmarking is not quite fair to my views. To add on to this point, in a recently online method called SPIN (Maher et al, 2023), Supp. Fig. 5b shows an example of NMF on smoothed data. The authors should address this issue and repeat the benchmarking.

Response 2.2:

We agree with you that NMF and other similar methods can also take spatially smoothed gene expression matrix as input. However, these methods do not incorporate the spatial smoothing step as part of their normal workflow. Moreover, there are many smoothing approaches available. For example, UTAG5 smooths the data by a summation of the cell/spot itself and its neighbours. CellCharter6 first uses variational autoencoders to map the input gene expression matrix into latent embeddings on which spatial smoothing is performed via concatenating the latent features of each cell with the features being averages of its neighbours at 1 to l steps away. Finally, SPIN7 smooths the input data by taking the average expression of each cell's randomly sub-sampled neighbours. At present, it is unclear which smoothing approach is the optimal for different input data and for each method. Furthermore, various smoothing techniques may have different impacts, whether positive or negative, on the downstream analyses. Such impacts are yet to be evaluated. To achieve a comprehensive benchmarking, we would need to test all the different smoothing methods and their various combinations with NMF and other methods, which we believe is out of scope of the current STAMP manuscript. As such, we respectfully disagree that the benchmarking presented in STAMP manuscript is not fair. To reduce the scope, we only consider NMF's performance on the SPIN smoothed data below.

Different from NMF and other competing methods, STAMP incorporates spatial smoothing into its framework. Moreover, its spatial smoothing approach is unique, and novel compared to existing ones. STAMP adopts a simplified graph convolution (SGC) to compute the spatially smoothed gene expression matrix. Simplified graph convolution, as a linear approximation of the convolution graph network, displays a notably improved efficiency. This is particularly critical when dealing with large spatial transcriptomic data. Most importantly, we have demonstrated in the manuscript that employing the combination of SGC smoothing and topic modelling in STAMP leads to favourable outcomes when analysing datasets obtained from diverse technology platforms and tissue types.

Here, we assessed NMF's performance on the Slide-seq V2 mouse brain dataset smoothed with the SPIN approach (Maher et al, 2023). Specifically, we normalized and scaled the input data, followed by subtraction of the minimum value to ensure positivity. The data was then smoothed with a random walk on the spatial adjacency matrix. We found that NMF was able to recover the CA1, CA2, CA3, and DG regions with the smoothed data (Figure R2.2.1). However, the magnitudes of the topic proportions were drastically altered to impact further analysis. For example, we found that the topics

25 – 29 had much larger values compared to the rest of the topics while not capturing regions of any biological significance.

Figure R2.2.1: Topic proportions recovered by NMF on SPIN-smoothed data.

We also further processed the NMF output into two formats, the binarized output where each spot was assigned to only one topic that bears the highest proportion, and clustering obtained from NMF's outputs (Figure R2.2.2). In the binarized output, we found that almost all spots were assigned to Topics 25 to 29, and in the clustered output, the correspondence of clusters to expected biological structures was lost. In contrast, both binarized and clustered outputs from STAMP topics produced more meaningful results (Supplementary Figure S1).

Figure R2.2.2 Binarized topic proportions and clustering of topic proportions returned by NMF.

Comment 2.3: For the two datasets mentioned above, I would also like to know how the authors get the label of tissue region annotation for each topic.

Response 2.3:

For the mouse hippocampus (Figure 2) and anterior/posterior brain (Figure 4) datasets, our annotations were derived manually by comparing the computed topics with the Allen brain atlas (also provided in the respective figures). We also used marker genes from literature to guide our annotations. This approach has also been previously employed in previous studies^{8,9}.

- In line 222-224, the authors found "STAMP-BC resolved the L5 layer into two cell type-specific topics." Any additional reference to validate the two cell types? What are some key marker genes?

Thank you very much for raising this question. After our initial submission, we investigated further on the cell types in the L5 layer. We plotted the gene expression of reported cell type markers¹⁰ and found that the cell types could not be clearly distinguished. Therefore, we hypothesized that the two L5 layers resulted from residual batch effects. With our continued efforts to improve STAMP, we also refined STAMP's batch correction capabilities. We ran the updated STAMP again on the data and the two topics are now combined into a single topic (Topic 8, L5) (revised Figure 4E). To confirm the new topic's identity, we took gene markers from the Allen Brain Atlas ISH data and checked their rankings within the new L5 topic's associated gene module (revised Figure 4F).

Figure R2.3.1: Spatial plot for markers genes for L5 IT

Figure R2.3.2: Spatial plot for markers genes for L5 PT

- In the mouse embryo development dataset (figure 5), the authors demonstrated "topics that capture the temporal dynamics of embryo development, including the brain and central nervous system (line 248-249)." There are a few similar temporal analysis workflows that use the same data (MOSCOT: Klein et al, 2023; STAligner: Zhou et al, 2022). How is the performance of STAMP compared to these methods? Do they capture similar spatiotemporal patterns?

STAligner is a graph attention network tool that can integrate multiple spatial transcriptomics samples and generate a latent embedding. However, we found that running STAligner on the full dataset of MOSTA requires a GPU with large amounts of memory available. We were unable to run it on our server with a GPU equipped with 12GB of memory. When running on CPU only, it takes too long for the full dataset with 500k cells. This is due to STAligner's usage of graph attention networks which cannot be mini-batched, and STAligner does not employ graph sampling techniques such as GraphSage and GraphSAINT which may help to circumvent the issue. Instead, we provide the results of STAligner with a down sampled dataset of 90k cells. We then used mclust, the default clustering algorithm recommended in STAligner, to cluster the embeddings into 40 clusters.

Figure R2.3.1 Clusters returned by STAligner.

Figure R2.3.2 Individual clusters returned by STAligner.

Here, we show the individual clusters separately for the ease of visualization. Many of STAligner's

clusters were similar to the topics returned by STAMP (Figure 6 and Supplementary Figure S25 of the revised manuscript). However, we also noticed that some of the STAligner clusters were not coherent compared to STAMP. For example, Cluster 13 (Heart) was not detected at the last time point E16.5. Cluster 11 (Liver) was also not properly identified.

For a quantitative comparison between STAMP and STAligner, we employed the scIB metrics. We used the time points as batch labels and the original annotation as the ground truth. We found that STAMP outperformed STAligner on this dataset (Figure R2.3.3) in terms of both batch-correction and bio-conservation, indicating STAMP's ability to identify more coherent topics across time.

Figure R2.3.3 scIB metrics computed for STAMP and STAligner.

MOSCOT is a toolbox that uses optimal transport for multiple different tasks. The task that is most relevant to STAMP is MOSCOT's spatial-temporal mapping module, which uses Gromov-Wasserstein optimal transport to find correspondences (transitions) between cells captured at two different time points. However, we believe that this module of MOSCOT is also not directly relevant to STAMP as it serves a different purpose where it seeks to evaluate the distances between cells with the use of an input embedding. Nevertheless, MOSCOT can take STAMP's topic proportions as input. Here we compared the results of running MOSCOT with STAMP's topic proportions and the default embedding (PCA). We found that STAMP enabled more accurate mapping across different time points (Figure R2.3.4, R2.3.5). Specifically, with STAMP's embedding, MOSCOT achieved more accurate alignment in multiple regions such as the jaw and tooth, lung, pancreas, brain, epidermis, and meninges region, while performed slightly poorer in the muscle region.

Figure R2.3.4 Transition probabilities returned by MOSCOT with STAMP embedding between time point E13.5 to E14.5

Figure R2.3.5 Transition probabilities returned by MOSCOT with default PCA embedding.

Comment 2.4: The original mouse organogenesis dataset (Chen et al, 2022) contains eight time points. However, it is unfortunate that the authors only showed the results from two of the time points in figure 5. To make the demonstration stronger, the authors should comment on why the two stages are selected and provide the spatiotemporal analysis (figure 5A-D) on more time points.

Response 2.4:

Thank you so much for this great suggestion. STAMP is capable of handling more than two time points. Initially, those two time points were simply chosen at random. However, we do agree with you on using all eight time points for a more comprehensive performance assessment. In our initial submission, STAMP has two modules, single-sample and multi-sample (i.e. STAMP-BC for batch correction), and we applied STAMP-BC to the mouse organogenesis dataset. With STAMP-BC, gene modules for the same topic representing an organ (or cell type) were shared across batches.

However, during the revision, we realized that for time series data, we should allow the gene modules to vary across different time points. As such, we have expanded STAMP with a third module to specifically analyse time series data. This new module allows the discovery of dynamic gene modules (i.e. time point specific gene modules). We applied the new STAMP time-series module to perform the spatiotemporal analysis of the eight different time points and captured organ/topics with dynamic gene modules that changed during mouse organogenesis (Figure 6 of the revised manuscript). Particularly, our analysis results showed that the associated gene modules reflected both the similarities within each topic between time points and the gradual changes across time that were aligned with development trajectories.

Comment 2.5: The authors "obtained the 50-binned mouse embryo (line 442)" for STAMP. I'm curious to know how the number of bins was chosen. In addition, what's the performance of STAMP on single-cell (or single spot)-resolved, non-binned data?

Response 2.5:

The mouse embryo dataset was acquired with Stereo-seq, which is built on the DNA Nanoball (DNB) technology. Standard DNB chips have spots with a diameter of approximately 220 nm, offering high spatial resolution. However, Stereo-seq data is highly sparse at the single spot level. As such, binning is typically necessary to address data sparsity and enable more meaningful downstream analyses. In the original study, the data was binned at 50 and the annotation was generated with the binned data 11. Here, we employed the same binned data without modification. For datasets acquired with other technologies, we applied STAMP to the single cell or spot resolved, non-binned data. For instance,

we applied STAMP to the Slide-seq mouse brain and SMI NSCLC data which were single cell resolved and non-binned. The analysis results were presented in Figures 2 and 3 of the revised manuscript.

Some minor comments:

Comment 2.6.1: "Covariates" were brought up in figure 1A, but no further description/explanation was provided in the manuscript. The authors should provide additional sentences to clarify the term.

Response 2.6.1:

We apologize for the confusion caused. Covariate generally refers to independent factors that are present but are not of interest to the analysis. Within the context of STAMP, the covariate factor being considered is the batch. Hence, to enhance clarity, we have removed the term covariates and used batches instead. We have also updated Figure 1 to better illustrate STAMP's methodology.

Comment 2.6.2: Can the authors provide explanations of the difference between gene topics and modules? It's particularly unclear to me what gene modules mean in the manuscript (line 92 and others).

Response 2.6.2:

Within STAMP, topics refer to the low dimensional embedding of cells (or spots). Each topic is associated with a gene module wherein genes are ranked (or scored) by their contribution to their respective topic. We have now provided a clearer explanation in Figure 1 and the revised Methods section. There we refer to Z as the topic proportion and W as the gene module. Each cell/spot can be assigned to more than one topic at varying proportions with the sum of proportions being equal to 1. The assignment of topic proportions is based on the decomposition of the gene expression by STAMP into the topics.

Comment 2.6.3: To make the paper more focused, I suggest that the authors can reduce or remove the discussions that are not STAMP (i.e. around the other competing methods).

Response 2.6.3:

We thank the reviewer for this suggestion. We discussed about competing methods within the introduction to give the reader a background on competing methods. In the other sections, we have now shortened or removed the discussion around the competing methods.

Comment 2.6.4: Texts in some figure panels are very hard to see. For example, in figure 1D and 4B.

Response 2.6.3:

We apologize on the issues with figure resolution resulting problems with text readability. We have regenerated the figures at a higher resolution to ensure readability.

Reviewer #3:

Remarks to the Author:

This manuscript applies a deep generative topic model for detecting latent spatial topics and gene modules in spatial transcriptomics data. The method is useful but the manuscript has a lot of clarity issues to be corrected.

We are happy that you find STAMP to be a useful method and we also appreciate your effort in reviewing the manuscript and spotting problems within. We have revised the manuscript based on comments from all three reviewers, as well as introduced further improvements to STAMP. Notably, STAMP can now analyse time-series data to capture topics and associated gene modules across tissue slices acquired at different time points. On a set of mouse embryo sections from different developmental stages, STAMP captured spatio-temporally linked topics and gene modules that modelled organ development across time, such as muscle and liver development.

We have also revised manuscript to improve its clarity and also reflect STAMP's new capabilities and included additional examples to demonstrate them. The benchmarks presented have been updated with more competing methods and quantitative metrics as well. Other updates include the Methods section and Figure 1 to better explain STAMP's methodology. We have also clarified your questions on STAMP'S methodology in the responses below. We are happy to answer additional queries on STAMP and the manuscript.

Comment 3.1: The graphical model in Figure 1C appears to have some errors. Specifically, c and λ are gene and topic specific. They should not be in the N-rectangle but in the G-rectangle.

Response 3.1:

Thank you for highlighting this error. We have revised the figure to rectify it. We have also updated the STAMP package which now has three modules for the analysis scenarios of single-sample analysis, multi-sample integration, and time series data analysis. For each module, we have created a new graphical model which is now presented in Supplementary Figure S31 (Supplementary Note 2) of the revised manuscript and also below.

Figure R3.1 Graphical models of the 3 different modes of STAMP. Scenario 1 refers to single sample STAMP, scenario 2 refers to multi sample STAMP, and scenario 3 refers to STAMP with time-series data. Note that in the third scenario, $wkg \in RT$, while in the first scenario, $wkg \in R1$. The length scale random variable of the Gaussian process is omitted in Scenario 3 as it is fixed to 1.

Comment 3.2: The pre-processing step to obtain the concatenated gene expression data matrix in Figure 1A is not clearly explained. The steps of building adjacency graph, normalization and smoothing with spatial graph and gene expression to obtain matrix TX should be explained. In addition, the batch effect features in S seems to be binary per experiment?

Response 3.2:

We agree that the pre-processing step descriptions can be expanded and improved. The spatial adjacency graphs are built with the k nearest neighbours identified based on the Euclidean distance using the `squidpy.pp.spatial_neighbors` function. The number of neighbours is set to 6 (equivalent to 1 hexagonal ring) for structured data (for example, 10x Genomics Visium) and 1/1000 of the total number of cells for other data. The detailed description of graph normalization and smoothing is provided in the Response 3.3 below. We have also revised the Methods section with this information to aid the reader in understanding STAMP's methodology.

In the original Figure 1, the batch effect features in S is binary with one-hot encoding, indicating which batch each cell belongs to. The number of columns in S is set to the number of actual data batches.

Comment 3.3: In addition, it is inaccurate to mention STAMP uses graph-convolutional neural network since the spatial information is only used to create the spatially smoothed gene expression as additional features in TX by a one-step random walk on the spatial graph (as shown in Figure 1A). This needs to be clarified.

Response 3.3:

We apologize for not being clear in our manuscript. To be more precise, STAMP uses a simplified graph convolutional (SGC) neural network (Wu, et al. and Frasca, et al.12,13), which is a linearization of the conventional graph-convolutional neural network (Kipf, et al.14). Here we describe the SGC architecture (which is also provided in the Supplementary Note 2 of the revised manuscript).

A l th layered graph convolutional network can be expressed as

$$\begin{aligned} \tilde{A} &= A + I \\ (0) \\ H &= X \\ 1 \ 1 \\ S &= \tilde{D}^{-2} \tilde{A} D \\ \sim -2 \\ H(l) &= \sigma(SH(l-1)W(l)) \end{aligned}$$

where X is the gene expression matrix and σ is a nonlinear activation function such as ReLU. I is the identity matrix. \tilde{A} is the degree of \tilde{A} . S is also known as the symmetric normalized adjacency matrix.

Wu et al.13 hypothesize that the success of GCNs are due to the local averaging and proposed a simplified graph convolution that linearizes the graph convolutional network by removing the nonlinear activation functions σ , which leads to

$$\begin{aligned} H(l) &= SH(l-1)W(l) \\ &= SSH(l-2)W(l-1)W(l) \\ &= S \dots SSH(0)W(1) \dots W(l) \\ &= S l X W \end{aligned}$$

where the weights $W(1) \dots W(l)$ can be combined into a single weight W .

The simplified graph convolution network is more efficient than the conventional graph convolutional neural network as it allows us to precompute the $S \cdot X$, which is equivalent to smoothing the gene expression with a normalized adjacency matrix. This aids in scalability, as we can now train the simple graph convolution network as a multi-layered perceptron with mini-batched training. Regular graph neural networks such as graph convolutional network and graph attention networks are unable to do so as we need to fit the whole adjacency graph in the GPU memory during training, therefore limiting their scalability to large datasets.

However, we observed that relying solely on the smoothed expression matrix results in over-smoothed latent topics. We hypothesize that this is due to their existing both spatial and non-spatial pattern in the dataset. Therefore, we concatenate the smoothed matrix with the original gene expression matrix. Note that the S here is not the batch label but the smoothing matrix.

Comment 3.4: The important measure "Topic proportion" shown in all the results is undefined. How is it calculated? It seems to be the gene column in Beta product with Z ?

Response 3.4:

Similar to PCA and NMF, STAMP is also a form of matrix factorization that decomposes the gene expression matrix into a product of low dimensional factors (i.e. cell embedding, termed topic proportion in STAMP) and gene embedding. The topic proportion is represented by Z and gene embedding by β in Figure 1. The topic proportions z_{nk} and gene module scores w_{kg} are taken to be the mean of the variational posterior $q(z_{nk} | X, A)$ and $q(w_{kg})$, respectively. We further post-process the gene module scores by downweighing the lowly-expressed genes. The new gene module score is given by:

$$w_{kg_new} = \log(r_g + \epsilon) - \log(r_g) + w_{kg},$$

where r_g is the mean of the posterior distribution and $q(r_g)$. We set ϵ to the 10th quantile of r_g .

Comment 3.5: It is not mentioned how the number of topics is decided on each dataset for each method.

Response 3.5:

STAMP allows users to select the number of topics through a data exploration process, experimenting with different topic numbers depending on the tissue complexity and desired granularity level. We offer the Evidence Lower Bound (ELBO) metric as a guiding criterion for the selection, which we found to be effective. Here, we illustrate it by presenting an example using the 10x Genomics Visium mouse brain dataset to showcase our selection of optimal topic number. Since ELBO is stochastic, we utilized locally weighted scatterplot smoothing (loess) to smooth out the ELBO values. We noticed that it converged at around 30 topics for the dataset, which led us to select 30 as the number of topics. However, we do believe that, similar to clustering, selecting the number of topics is also dependent on the user's requirements in terms of granularity for the analysis. Increasing the number of topics enhances the granularity of topics and likewise increases the resolution in uncovering cell types or structures, subject to data quality. This has been demonstrated in our response to Comment 1.7.

Figure R3.5.1 Plot of ELBO with respect to number of topics for the 10x Genomics Visium mouse brain dataset.

Indeed, users have the flexibility to increase the number of topics for higher granularity. With 15 topics, the T CD4 memory, T CD4 Naïve, and T CD 8 Naïve cells from the original annotation presented in Figure 3A were not identified by any STAMP topics. When we increased the number of topics to 20 (Figure R3.5.2), we uncovered the T/NK sub-populations, namely Topic 6 associated with CD4 T cell markers (IL7R, CD2), and Topic 13 associated with CD8 T or NK cell markers (CCL5, KLRK1) (Figure R3.5.3). These two topics were previously combined when STAMP was set to run for 15 topics. We also compared STAMP's topic proportions against the original annotation (Figure R3.5.5) and found that Topic 6 aligned with the annotated CD4 T and Topic 13 with the annotated CD8 T and NK cells, confirming their cell type identities of. This showed that with an increased number of topics, STAMP was able to reveal the subsets of T and NK cells. Interestingly, STAMP also revealed two additional topics (Topics 4 and 15) related to plasmablast subsets (Figure R3.5.4) that were absent in the original annotation. The top genes associated with Topic 4 pertain to the IgG heavy chain while highly ranked genes of Topic 15 included the IgA heavy chain, suggesting the presence of different

plasmablast subtypes.

Figure R3.5.2 Topic proportions generated by STAMP for 20 topics on the SMI NSCLC data.

Figure R3.5.3 Topic proportions and expression of top two genes for Topics 6 and 13 returned by STAMP for 20 topics.

Figure R3.5.4 Topic proportions of Topics 4 and 15 and expression of their respective top two genes returned by STAMP for 20 topics.

Figure R3.5.5 Comparison of STAMP's topics against the original annotations. Here colour represents topic proportion.

Comment 3.6: The details (such as the exact number of spots and genes) of each dataset should be listed in a table in supplementary document.

Response 3.6: Thank you for this very useful suggestion. We have now provided the details of each dataset in Table S2 in the Supplementary Materials and as follows.

Figure	Technology	Tissue	cells	genes	cells (pre-processed)	genes (pre-processed)
Slideseq	Mouse		39220	6000		
2A			41770	23264		
V2		brain				
3A	Spatial	Human	98002	960	93206	600
Molecular small-Imager		lung cell cancer				
Mouse			3289	32285		
brain						
Visium anterior						
4A			6110	2000		
Mouse			32285			
brain			2825			
Posterior						
Stereo-seq			107416	26145		
Mouse						
Visium			1185	32835	148047	
5A		olfactory	6000			
Slide-seq						
bulb			47249	22167		
V2						
Human			47329			
5D	Visium		47681	33538	4000	
DLPFC						
Mouse						
embryo			510312			
6A	Stereo-seq		520815	23761	2000	
E9.5						
E16.5						

Table R3.6: Detailed information of the datasets used.

Comment 3.7: In the definition of STAMP between line 314-321, the L_n term on line 314 is unexplained. It seems to be the diagonal entries of L_n or $L_n L_n^T$ introduced later on line 379? The notation needs to be improved for clarity here.

Response 3.7: We apologize for the missing variable definitions. In line 314, l_n refers to the library size of cell n . We have clarified this in the revised manuscript. We modelled the library size as the

observed expression sum of the cell as

$$ln = \sum_g xng$$

The other L_n refers to the lower triangular matrix of the Cholesky factorization, also known as the Cholesky factor, which is an output of the inference network. The covariance term can be constructed through $L_n L_n^T$. However, we have removed the Cholesky factor from our algorithm as it does not improve results while potentially leading to numerical errors. Instead, we parameterise the prior of the topic proportion with a covariance matrix in the form of $(UU^T + \sigma I)$

$$\begin{aligned} u_k &\sim \text{Normal}(\mathbf{0}, I) \\ \sigma &\sim \text{HalfCauchy}(1) \\ \tilde{z}_n &\sim \text{Normal}(\mathbf{0}, UU^T + \sigma I) \\ z_n &= \text{softmax}_k(\tilde{z}_n) \end{aligned}$$

where I is an identity matrix, $U = (u_1, u_2, \dots, u_k)$

Comment 3.8: On line 415, how does NPMI give measures of topic diversity and topic coherence with one ratio? Are they the nominator and the denominator?

Response 3.8: We apologize for the confusion on line 415 due to the language used. In brief, NPMI is used to compute coherence only and is not related to diversity. For greater clarity, here we describe in detail how topic diversity and topic coherence (now renamed as module diversity and module coherence in the revised manuscript) are calculated.

To evaluate the gene modules found, we used two metrics popular in topic modelling. The first is module coherence, which we measure with normalized pointwise mutual information (NPMI). NPMI quantifies the co-occurrence of genes, and the module coherence calculated by taking the mean.

$$\begin{aligned} &p(g_i, g_j) \\ &\log \\ &\frac{1}{20} \frac{1}{K} \frac{p(g_i, g_j)}{p(g_i)p(g_j)} \\ &Module\ Coherence = \frac{1}{K} \sum_{k=1}^{190} \sum_{i=1}^{20} \sum_{j=i+1}^{20} \log \frac{p(g_i, g_j)}{p(g_i)p(g_j)} \end{aligned}$$

where $p(g_i, g_j)$ refers to the joint probability of two genes occurring in the same cell, $p(g_i)$ refers to the probability of observing gene i and K is the number of topics. We use the observed counts of each gene to measure the probabilities. To prevent housekeeping or background genes from dominating the metric, we only consider a gene if its expression is in the top 25th quantile. We used the top 20 genes for our evaluation. The unnormalized version of the metric has also been used in other single cell studies^{15,16}.

The second is module diversity that measures how unique gene modules are using ranked bias overlap (RBO). RBO compares two gene modules M_i and M_j of equivalent size and returns a number between 0 and 1, where 1 means the two gene modules are identical and 0 means they are completely unique. We calculate RBO scores between each pair of gene modules using only the top 20 genes. The module diversity score is then calculated by taking the mean of the minimum $1 - RBO$ score per topic. This ensures that modules with a higher diversity obtains a higher score.

$$\begin{aligned} &1 \\ &Module\ Diversity = \frac{1}{K} \sum_{i=1}^{190} \min(1 - RBO(M_i, M_j) \text{ for } j \in K) \end{aligned}$$

References

1. Luecken, M. D. et al. Benchmarking atlas-level data integration in single-cell genomics. *Nat Methods* 19, 41–50 (2022).

2. Carbonetto, P., Sarkar, A., Wang, Z. & Stephens, M. Non-negative matrix factorization algorithms greatly improve topic model fits. Preprint at <https://doi.org/10.48550/arXiv.2105.13440> (2022).
3. Hamilton, W., Ying, Z. & Leskovec, J. Inductive Representation Learning on Large Graphs. in *Advances in Neural Information Processing Systems* vol. 30 (Curran Associates, Inc., 2017).
4. Zeng, H., Zhou, H., Srivastava, A., Kannan, R. & Prasanna, V. GraphSAINT: Graph Sampling Based Inductive Learning Method. Preprint at <https://doi.org/10.48550/arXiv.1907.04931> (2020).
5. Kim, J. et al. Unsupervised discovery of tissue architecture in multiplexed imaging. *Nat Methods* 19, 1653–1661 (2022).
6. Varrone, M., Tavernari, D., Santamaria-Martínez, A., Walsh, L. A. & Ciriello, G. CellCharter reveals spatial cell niches associated with tissue remodeling and cell plasticity. *Nat Genet* 56, 74–84 (2024).
7. Maher, K. et al. Mitigating autocorrelation during spatially resolved transcriptomics data analysis. 2023.06.30.547258 Preprint at <https://doi.org/10.1101/2023.06.30.547258> (2023).
8. Long, Y. et al. Spatially informed clustering, integration, and deconvolution of spatial transcriptomics with GraphST. *Nat Commun* 14, 1155 (2023).
9. Hu, J. et al. SpaGCN: Integrating gene expression, spatial location and histology to identify spatial domains and spatially variable genes by graph convolutional network. *Nat Methods* 18, 1342–1351 (2021).
10. Economo, M. N. et al. Distinct descending motor cortex pathways and their roles in movement. *Nature* 563, 79–84 (2018).
11. Chen, A. et al. Spatiotemporal transcriptomic atlas of mouse organogenesis using DNA nanoball-patterned arrays. *Cell* 185, 1777-1792.e21 (2022).
12. Frasca, F. et al. SIGN: Scalable Inception Graph Neural Networks. Preprint at <https://doi.org/10.48550/arXiv.2004.11198> (2020).
13. Wu, F. et al. Simplifying Graph Convolutional Networks. in *Proceedings of the 36th International Conference on Machine Learning* 6861–6871 (PMLR, 2019).
14. Kipf, T. N. & Welling, M. Semi-Supervised Classification with Graph Convolutional Networks. Preprint at <https://doi.org/10.48550/arXiv.1609.02907> (2017).
15. Kunes, R. Z., Walle, T., Nawy, T. & Pe'er, D. Supervised Discovery of Interpretable Gene Programs from Single-Cell Data. <http://biorxiv.org/lookup/doi/10.1101/2022.12.20.521311> (2022) doi:10.1101/2022.12.20.521311.
16. González-Blas, C. B. et al. SCENIC+: single-cell multiomic inference of enhancers and gene regulatory networks. 2022.08.19.504505 Preprint at <https://doi.org/10.1101/2022.08.19.504505> (2022).

Decision Letter:

Our ref: NMETH-A53409A

17th Jun 2024

Dear Jinmiao,

Thank you for submitting your revised manuscript "Interpretable spatially aware dimension reduction of spatial transcriptomics with STAMP" (NMETH-A53409A). It has now been seen by the original referees and their comments are below. We also sent your second rebuttal to reviewer 2, who was satisfied with your new data and proposed changes.

The reviewers find that the paper has improved in revision, and therefore we'll be happy in principle to publish it in Nature Methods, pending minor revisions to satisfy the referees' final requests and to comply with our editorial and formatting guidelines. We ask that you revise the manuscript as described and resubmit with the point-by-point rebuttal.

TRANSPARENT PEER REVIEW

Please note: we allow redactions to authors' rebuttal and reviewer comments in the interest of confidentiality. If you are concerned about the release of confidential data, please let us know specifically what information you would like to have removed. Please note that we cannot incorporate redactions for any other reasons. Reviewer names will be published in the peer review files if the reviewer signed the comments to authors, or if reviewers explicitly agree to release their name. For more information, please refer to our <https://www.nature.com/documents/nr-transparent-peer-review.pdf> target="new">FAQ page.

ORCID

Sincerely,
Rita

Rita Strack, Ph.D.
Senior Editor
Nature Methods

Reviewer #1 (Remarks to the Author):

The authors have thoroughly addressed my comments, and it looks much improved now.

Reviewer #1 (Remarks on code availability):

We could install the tool and run some example tutorials they provided.

Reviewer #2 (Remarks to the Author):

The revised manuscript addressed many important technical concerns, including extending the spatiotemporal analysis to 8 time points for the embryogenesis dataset and adding more benchmarking with other methods. I'm glad to see that the authors have already enhanced STAMP by adding analysis for time series data. I would also like to thank the authors for

their detailed clarifications regarding the key advantage of STAMP compared to other methods. Overall, I think the paper is much improved and the method will be of great interest to the field. Yet, I still have two main questions at this point:

1. Considering the neighborhood information is widely used strategy in spatial transcriptome, I initially thought simply applying NMF and other methods like LDA and LDVAE for gene expression matrix is not fair. To claim the strength of the STAMP, I also suggest authors could benchmark with these methods on the smoothed data from the simplified graph convolution which STAMP adopts.

Meanwhile, it seems that the NMF on smoothed gene expression can also well identified the selected topics from STAMP in Fig 2E (such as topic8-CA3, topic9-CA1, topic24-CA2, topic3-DG, as well as several clear layer topics). And the CA2 topic looks even better than the result in the manuscript. Can the authors provide more emphasis on what's more we can get from STAMP? I suggest that the authors can discuss in more details about the novelty and values of the STAMP in the manuscript.

2. The initial region labels of stereo-seq embryo samples are generated by the Dynamo with function `[dynamo.tl](http://dynamo.tl/.scc)`, which considered both gene expression and neighbor gene expression. Since one of the key points made by the authors is that it is interpretable, I am curious about what additional or improved outcomes STAMP can produce in this case? It also looks to me that the clusters generated by STAMP have many discontinuous noise within the regions. Can the model generate clear tissue regions? Since there are so many region segmentation tools are available, I suggest the authors provide more evidence that the method has best performance.

Reviewer #3 (Remarks to the Author):

The authors have addressed the issues raised in the previous review. I have no further comments.

Author Rebuttal letter:

We appreciate the comments made by the reviewer to strengthen the manuscript. We hope the responses below address the concerns expressed.

Comment: Considering the neighborhood information is widely used strategy in spatial transcriptome, I initially thought simply applying NMF and other methods like LDA and LDVAE for gene expression matrix is not fair. To claim the strength of the STAMP, I also suggest authors could benchmark with these methods on the smoothed data from the simplified graph convolution which STAMP adopts.

Response: We understand the benchmarking idea suggested here by the reviewer. However, there are technical aspects that pose feasibility problems. Simplified graph convolution (SGC) as used in STAMP is a linear approximation of the graph convolutional network, which is mathematically equivalent to normalized adjacency smoothing followed by passing through a neural network. Therefore, our SGC implementation cannot be simply used in conjunction with LDA and NMF. It is possible to use SGC with LDVAE, which is a deep learning-based method, but this will require architectural changes to be made to LDVAE. We believe making such changes to other methods is out of the scope of our paper.

For a more reasonable testing regime, we applied normalized adjacency smoothing without passing through a neural network to produce smoothed data as input to the competing methods. The benchmarking results of NMF, LDA and LDVAE are presented below. Visually, NMF (Figure R1) and LDA's (Figure R2) results were significantly worse than STAMP's results. LDVAE (Figure R3) now captured the LH and separated the V3 and MH, but CA2 remained merged with CA3. Furthermore, it appears that most of the results were oversmoothed and that there were many disjoint topics produced. We next qualitatively benchmarked the associated gene modules. In terms of gene module coherence and diversity, the performance of all three methods were degraded compared to using the unsmoothed data (Figure R4). Overall, the results demonstrated our point that the choice of smoothing algorithm is not an easy one; there is no universal optimal smoothing algorithm. It also illustrated that SGC was not the sole reason behind the superior performance of STAMP on this dataset.

Figure R1: NMF results with SGC smoothed data

Figure R2: LDA results with SGC smoothed data

Figure R3: LDVAE results with SGC smoothed data

Figure R4: Quantitative gene module metrics

Comment: Meanwhile, it seems that the NMF on smoothed gene expression can also well identified the selected topics from STAMP in Fig 2E (such as topic8-CA3, topic9-CA1, topic24-CA2, topic3-DG, as well as several clear layer topics). And the CA2 topic looks even better than the result in the manuscript. Can the authors provide more emphasis on what's more we can get from STAMP? I suggest that the authors can discuss in more details about the novelty and values of the STAMP in the manuscript.

Response: We agree with the reviewer that the results of NMF with the SPIN-smoothed gene expression also clearly demarcated the CA3, CA2, CA1, DG, and other regions. However, a

side effect of this smoothing is that the magnitude of the topic proportions was disrupted in the process. Notably, the topics 27 – 29 had much larger values (around 0.8) compared to the rest of the topics. This can potentially give a false impression of those topics being highly important and may lead to misinterpretation, particularly in the absence of prior knowledge about the tissue being studied. Moreover, in terms of quantitative metrics, SPIN-NMF scored lower than STAMP (Figure R4) for both the module coherence and module diversity metrics. To further demonstrate the impact of magnitude distortion, we ran SPIN-NMF on a simulation dataset presented in Supplementary Note 1: Simulation study of the revised manuscript. We found that SPIN-NMF was unable to recover the correct underlying patterns and their magnitudes (Figure R5).

a

b

Figure R5: Ground truth patterns (Panel a), ground truth topic proportions (Panel b), and proportions returned by SPIN-NMF (bottom).

We would also like to highlight that while there are combinations of methods and data preprocessing such as NMF with SPIN that is competitive with STAMP on specific datasets, we have shown that STAMP offers superior performance in other scenarios. Specifically, STAMP outperformed NMF, LDA, and LDVAE on the tasks of integrating adjacent tissue sections (Figure 4, revised manuscript) and batch integration of data generated with different technology platforms (Figure 5, revised manuscript). STAMP also offers functionality not available with other methods. Specifically, STAMP is the first topic modelling method that can capture shared topics across time-series spatial transcriptomic data. With the mouse embryonic development dataset, STAMP revealed spatio-temporally linked topics and their associated gene modules (Figure 6, revised manuscript). These topics matched known biological structures and the associated gene modules tracked changes in the contributing genes across time. In our further analysis, we found topics that matched phenotypes like hematopoiesis and hepatocyte development. These topics not only varied spatio-temporally but also overlapped at certain time points; STAMP was able to successfully disentangle them.

Comment: The initial region labels of stereo-seq embryo samples are generated by the Dynamo with function `dynamo.tl.scc``, which considered both gene expression and neighbor gene expression. Since one of the key points made by the authors is that it is interpretable, I am curious about what additional or improved outcomes STAMP can produce in this case? It also looks to me that the clusters generated by STAMP have many discontinuous noise within the regions. Can the model generate clear tissue regions? Since there are so many region segmentation tools are available, I suggest the authors provide more evidence that the method has best performance.

Response: First, we would like to emphasize that STAMP is a dimension reduction method, rather than a clustering method. It identifies low dimensional topics and assigns a gene module to each topic. As such, its topics possess interpretability from the associated gene modules, which can be an alternative to clustering and DEG analysis. Furthermore, different from the typical clustering analysis that assigns a cell/spot to only one cluster, topic modelling gives a probabilistic topic composition to each cell/spot. The latter is a form of soft clustering that allows each cell/spot to possess multiple identities (i.e. topics). Furthermore, we believe that STAMP returns both spatial and non-spatial topics. Some of the topics, especially non-spatial ones that represent cell types present across different regions, are likely to be more discontinuous. As a result, when we assigned each cell/spot to the STAMP topic that presented the highest proportion score, the “clusters” in Figure 6B show some discontinuous noise. Moreover, in the results presented within the manuscript, achieving optimal spatial smoothness had not been a major design goal for STAMP. However, the spatial smoothness of STAMP’s topics can be controlled through hyperparameter tuning of the number of SGC’s layers, where a higher number of layers returns smoother topics. Here we show the results obtained with the embryo section at E11.5 (Figure R6). We further used Moran I’s score to quantitatively demonstrate the changes in the topics’ spatial smoothness (Table R1). Users can adjust this parameter in STAMP to obtain their preferred level of spatial smoothness. This feature offers flexibility and avoids spatial over-smoothing, particularly in studies where non-spatial topics and features are highly relevant.

As an interpretable dimension reduction method, STAMP has other unique advantages over clustering analysis. STAMP’s soft clustering enables it to capture multiple phenotypes that can spatially overlap. This is not possible with typical clustering approaches. With a

simulation dataset (Supplementary Note 1: Simulation study), we demonstrated that clustering methods like STAGATE and GraphST can't identify spatially overlapping patterns. Moreover, with the time-series mouse embryo data (Figure 6F-H of the revised manuscript), STAMP disentangled the developmental trajectories of erythro-myeloid hematopoiesis and hepatocytes within the liver during mouse embryo development. Therefore, we believe that the value of STAMP as a method is not solely illustrated by clustering metrics.

Lastly, regarding the request for benchmarking STAMP against region segmentation tools (i.e. clustering tools), we would like to highlight that STAMP dimension reduction and clustering are not mutually exclusive. STAMP outputs can be used as input for follow-up clustering and other downstream analyses. As demonstrated in Figure 5 of the revised manuscript, we employed the output of STAMP in k-means clustering and compared the results with those of region segmentation methods such as PRECAST, STAligner, and DeepST on datasets with known ground truths. In the tests, STAMP performed favorably compared to the other methods. Although this benchmarking was performed only on multi-sample integration tasks, we believe that a more thorough benchmarking of STAMP against clustering methods may not add much value to the manuscript but instead distract the readers from the focus of our study.

Figure R6: Binarized topics with hyperparameter tuning of number of SGC layers.

Number of SGC layers Moran's I score

1 0.795

2 0.856

3 0.873

4 0.877

Table R1: Moran's I score of topics obtained with different numbers of SGC layers.
